# Neural Sculpting: Uncovering hierarchically modular task structure in neural networks through pruning and network analysis

**Shreyas Malakarjun Patil**[1]   **Loizos Michael**[2,3]   **Constantine Dovrolis**[4,1]

[1]Georgia Institute of Technology, Atlanta, USA    [2]Open University of Cyprus, Nicosia, Cyprus

[3]CYENS Center of Excellence, Nicosia, Cyprus    [4]The Cyprus Institute, Nicosia, Cyprus

sm_patil@gatech.edu, loizos@ouc.ac.cy, c.dovrolis@cyi.ac.cy

## Abstract

Natural target functions and tasks typically exhibit hierarchical modularity – they can be broken down into simpler sub-functions that are organized in a hierarchy. Such sub-functions have two important features: they have a distinct set of inputs (*input-separability*) and they are reused as inputs higher in the hierarchy (*reusability*). Previous studies have established that hierarchically modular neural networks, which are inherently sparse, offer benefits such as learning efficiency, generalization, multi-task learning, and transfer. However, identifying the underlying sub-functions and their hierarchical structure for a given task can be challenging. The high-level question in this work is: if we learn a task using a sufficiently deep neural network, how can we uncover the underlying hierarchy of sub-functions in that task? As a starting point, we examine the domain of Boolean functions, where it is easier to determine whether a task is hierarchically modular. We propose an approach based on iterative unit and edge pruning (during training), combined with network analysis for module detection and hierarchy inference. Finally, we demonstrate that this method can uncover the hierarchical modularity of a wide range of Boolean functions and two vision tasks based on the MNIST digits dataset.

## 1   Introduction

Modular tasks typically consist of smaller sub-functions that operate on distinct input modalities, such as visual, auditory, or haptic inputs. Additionally, modular tasks are often hierarchical, with simpler sub-functions embedded in, or reused by, more complex functions [1]. Consequently, hierarchical modularity is a key organizing principle studied in both engineering and biological systems. In neuroscience, the hierarchical modularity of the brain's neural circuits is believed to play a crucial role in its ability to process information efficiently and adaptively [2, 3, 4, 5]. Translating this hierarchical modularity to artificial neural networks (NNs) can potentially lead to more efficient, adaptable and interpretable learning systems. Prior works have already shown that modular NNs efficiently adapt to new tasks [6, 7, 8] and display superior generalization over standard NNs [9, 10, 11]. However, those studies assume knowledge of the task's hierarchy, or hand-design, modular NNs at initialization. Given an arbitrary task, however, we normally do not know its underlying sub-functions or their hierarchical organization. The high-level question in this work is: *If we learn a task using a sufficiently deep NN, how can we uncover the underlying hierarchical organization of sub-functions in that task?*

Recent studies through NN unit clustering have demonstrated that certain modular structures can emerge during the training of NNs [12, 13, 14, 15, 16]. However, it is unclear whether the structures extracted reflect the underlying hierarchy of sub-functions in a task. Csordas et al. [17] proposed a method that identifies sub-networks in NNs that learn specific sub-functions. This method however,

37th Conference on Neural Information Processing Systems (NeurIPS 2023).

requires knowledge of the exact sub-functions within a task's hierarchy. Ideally, modules corresponding to specific sub-functions should emerge through a training strategy, and a method should be available to detect these modules without explicit knowledge of the corresponding sub-functions.

Biological networks exhibit hierarchical modularity where clusters of nodes with relatively dense internal connectivity and sparse external connectivity learn specific sub-functions [4, 5, 18, 19]. Drawing inspiration from this, we propose *Neural Sculpting*, a novel approach to train and structurally organize NN units to reveal the underlying hierarchy of sub-functions in a task. Within a hierarchically modular task, we consider sub-functions that are input-separable and reused. We first show that conventionally trained NNs do not acquire structural properties that reflect the previous sub-function properties. To address this, we introduce a sequential unit and edge pruning method to train NNs. Unit pruning first conditions the NN to learn reused sub-functions, while edge pruning subsequently reveals the sparse connectivity between the learned sub-functions. Finally, we propose a network analysis tool to uncover modules and their hierarchical organization within the sparse NNs.

We demonstrate the capability of *Neural Sculpting* to reveal the structure of diverse hierarchically modular Boolean tasks and two MNIST digits based tasks. To the best of our knowledge, this paper is the first to analyze specific sub-function properties within a hierarchically modular task and propose an end-to-end methodology to uncover its structure. This work also sheds light on the potential of pruning methods to uncover and harness structural properties in NNs.

## 1.1 Preliminary

To represent the decomposition of a task, we visualize it as a graph. Initially, we consider Boolean functions and their corresponding function graphs. A Boolean function $f : \{0,1\}^n \to \{0,1\}^m$, maps $n$ input bits to $m$ output bits. We define the set of gates $G$ as $\{\wedge, \vee, \mathbb{I}\}$, where $\wedge$ represents logical conjunction (AND), $\vee$ represents logical disjunction (OR), and $\mathbb{I}$ represents the identity function (ID). Additionally, we define the set of edge-types $E$ as $\{\to, \neg\}$, where $\to$ represents a transfer edge and $\neg$ represents a negation edge.

**Function Graph:** A $(G, E)$-***graph*** representing a Boolean function $f : \{0,1\}^n \to \{0,1\}^m$ is a directed acyclic graph comprising: $n$ sequentially ordered vertices with zero in-degree, designated as the ***input nodes***; $k$ vertices with non-zero in-degree, designated as the ***gate nodes***, with each vertex associated with a gate in $G$ and each of its in-edges associated with an edge-type in $E$; and $m$ sequentially-ordered vertices with zero out-degree, designated as the ***output nodes***.

We use the $G$ and $E$ defined above, as NN units have been demonstrated to learn these universal gates [20]. Function graphs, which break down complex computations into simpler ones, are typically sparse. Modularity in sparse graphs refers to the structural organization of subsets of nodes that exhibit strong internal and weak external connectivity, which are grouped into modules.

**Sub-function:** A *sub-function* is a subset of nodes within the function graph that collectively perform a specific task. The sub-function is characterized by having strong internal connectivity within its nodes, meaning that they are highly interdependent and work together to achieve the desired output. At the same time, nodes in the sub-function have weak external connectivity, indicating that they are relatively independent from the rest of the graph.

**Input Separable:** Two sub-functions are considered *input separable* if their in-edges originate from distinct and non-overlapping subsets of nodes in the function graph.

**Reused:** A sub-function is *reused* if it has two or more outgoing edges in the function graph. [1]

**Training NNs on target Boolean functions:** We start by obtaining the truth table of each function graph, which serves as our data source. The training set $(\boldsymbol{\mathcal{X}}_t, \boldsymbol{\mathcal{Y}}_t)$ includes the complete truth table with added random noise ($\mathcal{N}(0, 0.1)$) during each iteration to increase the number of training samples. The validation set $(\boldsymbol{\mathcal{X}}_v, \boldsymbol{\mathcal{Y}}_v)$ consists of the noise-free rows of the truth table. We use multi-layered perceptrons (MLPs) with ReLU activation for the hidden units and Kaiming weight initialization to learn the Boolean functions. The loss function is bitwise cross-entropy with Sigmoid activation. The NNs are trained using Adam optimizer with L2 regularization of $1e - 4$.

---

[1]Distinction between function reuse and operation reuse. Consider the AND gate which is an operation that is independent of its input variables. On the other hand, a sub-function is a combination of such operations along with specific inputs. Specific operations have to be relearned if applied to different inputs due to the fixed data flow in NNs [17].

## 2 Standard training of NNs is not enough

We show that NNs through standard training, do not acquire structural properties that reflect the properties of sub-functions. Specifically, we consider the two properties in isolation by constructing two graphs: one with input separable sub-functions, and the other with a reused sub-function.

### 2.1 Input Separable

The first function graph we consider has 4 input nodes and 4 output nodes. The output nodes $\{y_1, y_2\}$ depend only on $\{x_1, x_2\}$, while the output nodes $\{y_3, y_4\}$ depend only on $\{x_3, x_4\}$ (Figure 1a). We train various NN architectures to learn the target function with perfect accuracy on the validation set. The property of the function graph that reflects input-separable sub-functions is that there are no paths from input nodes $\{x_1, x_2\}$ to output nodes $\{y_3, y_4\}$ and from input nodes $\{x_3, x_4\}$ to output nodes $\{y_1, y_2\}$. However, in a neural network, all input units are connected to all output units

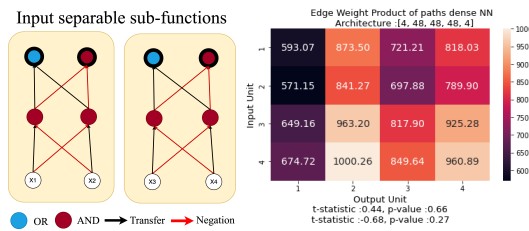

Figure 1: a. Function graph with input separable sub-function, b. Edge-weight product of paths from input units to output units in a trained NN.

through the same number of paths. Therefore, we analyze the strength of their learned relationship by considering the product of weight magnitudes along those paths.

Consider a trained NN parametrized by $\boldsymbol{\theta} \in \mathbb{R}^a$ and the set of all paths, $\mathcal{P}$. Each edge weight $\theta_i$ is assigned a binary variable $p_i$ to indicate whether it belongs to a given path $p \in \mathcal{P}$. We define the edge-weight product of a path as $\boldsymbol{\pi}_p = \prod_{i=1}^a |\theta_i^{p_i}|$, and $\boldsymbol{\pi}(i, j) = \sum_{p \in \mathcal{P}_{i \to j}} \boldsymbol{\pi}_p$ represents the sum of $\boldsymbol{\pi}_p$ for all paths from input unit $i$ to output unit $j$. We evaluate $\boldsymbol{\pi}(i, j)$ for $i = 1, 2$ and $j = 3, 4$, and $\boldsymbol{\pi}(i, j)$ for $i = 1, 2$ and $j = 1, 2$. If the former is significantly smaller than the latter, then the input units 1 and 2 are not used by output units 3 and 4.

| Width | 24 | 36 | 48 |
|---|---|---|---|
| Hidden Layers | | | |
| 1 | ✓ | ✓ | ✓ |
| 2 | ✗ | ✗ | ✗ |
| 3 | ✗ | ✗ | ✗ |

Table 1: NN architectures with varying widths and depths for which both null hypotheses are rejected.

We perform a two-sample mean test with unknown standard deviations, with $(\mu_1, s_1)$ representing the mean and sample standard deviation of $\boldsymbol{\pi}(i, j)$ for $i = 1, 2$ and $j = 3, 4$, and $(\mu_2, s_2)$ representing the mean and sample standard deviation of $\boldsymbol{\pi}(i, j)$ for $i = 1, 2$ and $j = 1, 2$. The null hypothesis $H_0$ is $\mu_1 = \mu_2$, and the alternative hypothesis is $\mu_1 < \mu_2$. A similar test is performed for input units 3 and 4. Figure 1b shows an example heat map for $\boldsymbol{\pi}(i, j)$, where $i, j \in \{1, 2, 3, 4\}$. For the example shown we cannot reject the null hypotheses. Similarly, we conducted statistical tests on nine different NNs with varying architectures and seed values, and the results are summarized in Table 1. For NNs with a single hidden layer, we can reject the null hypotheses, but for deeper NNs, both the null hypotheses cannot be rejected.

### 2.2 Reused

Consider the function graph shown in Figure 2a, consisting of 4 input nodes and 16 output nodes. First, we construct an intermediate sub-function $g(X)$ such that the three gate nodes depend on all the inputs. This sub-function is utilized by two separate gates, $f_1(g)$ and $f_2(g)$, which are then used 8 times by different output nodes. All paths to the output nodes pass through three gate nodes in the first hierarchical level and two gate nodes in the second. We analyze the edge-weight product of the paths from hidden units to output units. Let $\pi_p^l$ denote the sum of $\boldsymbol{\pi}_p$ for all paths originating from hidden layer $l$, which contains $N_l$ hidden units. We compute

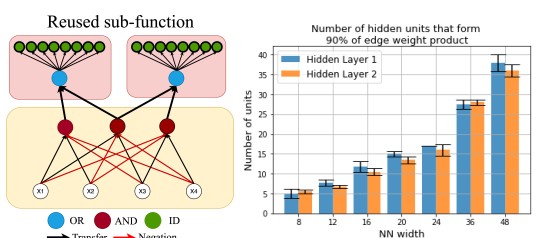

Figure 2: a. Function graph with reused sub-function, b. the number of units covering 90% of the total edge-weight product of paths.

the minimum number of units, $N_P^l$, that are necessary to achieve $P\%$ of the total $\boldsymbol{\pi}_p$. By comparing $N_P^l$ to that of the function graph, we can conclude whether the NN has learned the reused states.

We independently trained NNs with two hidden layers and different widths on the target Boolean function. The resulting $N_{90}^l$ values for different layers are shown in Figure 2b. Our analysis indicates that, as the width of the NN increases, so does $N_{90}^l$ in both the hidden layers, and these values are consistently close to the actual width of the NN. We trained an NN with hidden layers of width 3 and 2, respectively, to confirm that NNs with those widths can learn the function well.

## 3 Iterative pruning of NNs

In the previous section, we demonstrated that standard NN training is inadequate for learning structural properties that reveal input separable or reused sub-functions. Nevertheless, we observed that NNs with a relatively low number of parameters could still acquire these properties. Since hierarchical modularity is a property of sparse networks, we explored NN pruning algorithms as a means of achieving this sparsity. Prior research has shown that pruning NNs can reduce their number of parameters without compromising their performance [21, 22, 23]. Edge pruning methods typically require specifying a target NN density or pruning ratio [21, 24], but determining a density value that preserves the NN's performance remains an open question. Recently, iterative magnitude pruning [25, 26] has emerged as a natural solution to this problem. The algorithm prunes edges iteratively, removing some edges in each iteration and then retraining the NN. This process can be repeated until the NN achieves the same validation performance as the dense NN while being as sparse as possible.

### 3.1 Iterative edge pruning

Consider the initial edge pruning step $p_e$. We train the dense NN and prune $p_e\%$ of the edges with the lowest weight magnitude. The sparse NN resulting from this pruning is then trained with the same number of epochs and learning rate schedule as the original NN. We repeat this process until the sparse NN can no longer achieve the same validation accuracy as the dense NN. At this point, we rewind the NN to the previous sparse NN that achieved the same validation accuracy as the dense NN and update the value of $p_e$ as $p_e = p_e/2$. We repeat this process until $p_e$ becomes lower than the required step to prune a single edge, thereby ensuring that the lowest possible density is achieved.

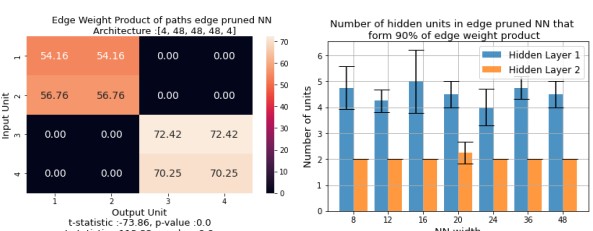

We next apply the tests developed in the previous section to analyze whether edge-pruned NNs acquire structural properties resembling those of the sub-functions. Specifically, we train and Figure 3: a. Edge-weight product of paths from input units to output units in edge-pruned NN; b. the number of units covering 90% of the total edge-weight product of paths in edge-pruned NNs.

prune different NN architectures to learn the function graph with input-separable sub-functions. For the example shown in Figure 3a, as well as for NNs with different architectures, we can reject the null hypotheses in favor of the alternate hypotheses. This implies that edge-pruned NNs acquire structural properties that enable them to recognize input-separable sub-functions.

Consider the previous function graph that has reused sub-functions and NNs with increasing widths. We independently train and prune each NN architecture and determine the number of units $N_{90}^l$ (Figure 3b). We find that in a majority of the trials, $N_{90}^2 = 2$, which are reused by the output units. Although there is a significant reduction in $N_{90}^1$, the edge-pruned NNs fail to identify 3 units that are being reused. We observe that edge-pruned NNs identify sparse connectivity between units and reuse those units. However, they do not identify units corresponding to sub-functions that are not sparsely connected yet reused. We hypothesize that this may be due to the initial pruning iterations where some edges from all units are removed, leaving no hidden units with dense connectivity to learn densely connected and reused sub-functions. Additionally, this could also be due to very large NN widths and the absence of an objective conditioning NNs to utilize as few hidden units as possible. Therefore, next we introduce an iterative unit pruning method to limit the number of units used.

## 3.2 Iterative hidden unit pruning

To prune hidden units, we first train the neural network and then assign each hidden unit a score. We eliminate $p_u\%$ of the hidden units with the lowest scores and train the resulting pruned network for the same number of epochs and learning rate schedule as the original network. We repeat this two-step pruning process iteratively.

We use loss sensitivity as the scoring metric for the units [27, 28, 29, 30], which approximates the change in the network's loss when the activation of a particular hidden unit is set to zero. Specifically, for a unit $i$ in hidden layer $l$, with activation $a_i^l$, the score is computed as :

$$S_i^l = \left| \frac{\partial \mathcal{L}(\mathcal{D}_v, \boldsymbol{\theta})}{\partial a_i^l} \times a_i^l \right| \quad (1)$$

The iterative unit pruning process continues until the NN can no longer achieve the same validation accuracy as the original NN. To ensure that we have pruned as many units as possible, we revert to the latest NN that achieved the same validation accuracy as the original NN and halve the value of $p_u$. This process is repeated until the unit pruning step becomes lower than the step size needed to prune a single unit. We perform unit pruning before the edge pruning process to identify the minimum widths required in each hidden layer. Once the minimum widths are determined, edges are pruned to reveal the sparse connectivity between those units. For additional details please refer to appendix section H.

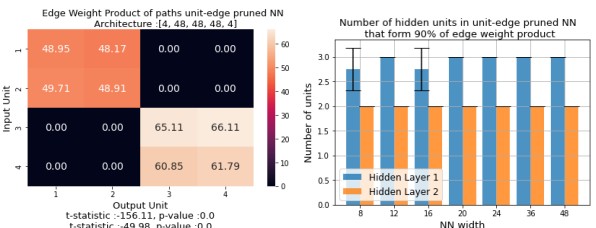

Figure 4: a. Edge-weight product of paths from input units to output units in unit-edge pruned NN; b. the number of units covering 90% of the total edge-weight product of paths in unit-edge pruned NNs.

We conducted the previous tests to analyze whether the unit-edge pruned NNs acquire structural properties resembling those of the sub-functions, as presented in the previous section. The results of these tests are shown in Figure 4, and show that the unit-edge pruned NNs do acquire structural properties resembling both input separable sub-functions and reused sub-functions.

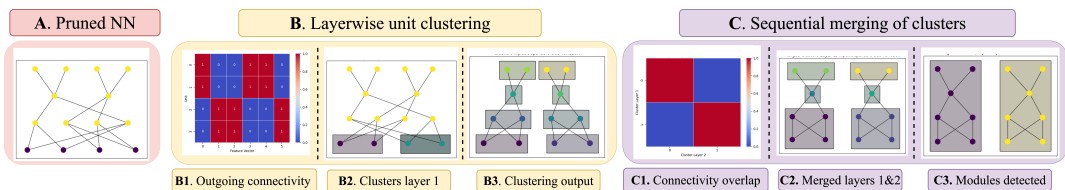

Figure 5: Proposed module detection pipeline.

## 4 Detecting modules within sparse NNs

In this section, we propose a method to uncover sub-networks or modules that approximate the sub-functions within the target function. We approach this by projecting the problem as a two-step partitioning problem due to the layered structure of NNs. First, we cluster the units belonging to the same layer. We assume that given a layer, there exist various subsets of units that participate in learning the same sub-function. Next, we merge the previous unit clusters across layers such that they exhibit strong connectivity. The overview of our proposed method is illustrated in Figure 5.

**Layer-wise unit clustering:** Let us consider a single layer $l$ with $N_l$ units. For each unit, we construct a feature vector based on its outgoing connectivity. The feature vector for a unit $i$ is a binary vector $f_i^l \in \{0, 1\}^g$, where $g$ is the total number of units in all the later layers. If unit $j$ is connected to unit $i$ through at least one directed path, then $f_i^l(j) = 1$, otherwise $f_i^l(j) = 0$. Our hypothesis is that the units that participate in learning the same sub-function are reused by similar units in the later layers. To partition the units into clusters such that their feature vectors have low intra-cluster and high inter-cluster distances, we use the Agglomerative clustering method with cosine distance and

average linkage. To identify the optimal number of clusters $K_l$, we use the modularity metric, which we have modified for our problem [31, 32, 33].

Let $\tilde{D}$ be the normalized distance matrix of the unit feature vectors, where the sum of all elements is equal to 1. Consider the units partitioned into $k$ clusters and a matrix $A \in \mathbb{R}^{k \times k}$, where $A_{ij} = \sum_{a \in C_i, b \in C_j} \tilde{D}_{ab}$ represents the sum of the distances between all pairs of units in clusters $C_i$ and $C_j$. The modularity metric, denoted by $M$, measures the quality of the partitioning and is defined as:

$$M = \sum_{i=1}^{k} \left( A_{ii} - \left[ \sum_{j=1}^{k} A_{ij} \right]^2 \right) \quad (2)$$

The first term in this equation measures the total intra-cluster distance between the unit features while the second term is the expected intra-cluster distance under the null hypothesis that unit distances were randomly assigned based on joint probabilities in $\tilde{D}$. A negative value of $M$ indicates that pair-wise distance within a cluster is lower than the random baseline. We iterate through values of $k$ ranging from 2 to $N_l - 1$, which are obtained through Agglomerative clustering, and select the value of $k$ that minimizes the modularity metric.

The modularity metric can accurately detect the presence of multiple unit clusters ($K_l = 2, ..., N_l - 1$). However, it fails for the edge cases where all units may belong to a single cluster ($K_l = 1$) or to separate individual clusters ($K_l = N_l$). To address those, we conduct a separability test if the modularity metric is lowest for $k = 2$ or $k = N_l - 1$, or if the modularity metric values are close to zero. Modularity metric close to zero is an indicator that the intra-cluster distances are not significantly different from the random baseline. To determine this, we set a threshold on the lowest modularity value obtained.

**Separability test:** The unit separability test is designed to evaluate whether two units in a cluster can be separated into sub-clusters. Consider two units $i$ and $j$, with $o_i = \sum f_i^l$, $o_j = \sum f_j^l$ neighbors respectively, and $o_{ij} = f_i^l \odot f_j^l$ common neighbors. We consider a random baseline that preserves $o_i$ and $o_j$. The number of common neighbors is modeled as a binomial random variable with $g$ trials and probability of success $p = \frac{o_i \times o_j}{g^2}$. The units are separable if the observed value of $o_{ij}$ is less than the expected value $\mathbb{E}(o_{ij})$ under the random model.

Consider the partition where $N_l - 1$ clusters are obtained. If the two units that are found in the same cluster are separable, it implies that all units belong to separate clusters. Now let us consider the partition of units into two clusters. We merge the feature vectors of the two unit groups. If the two groups of units are not separable, it implies that all units must belong to the same cluster. In some cases, both tests yield positive results. We determine the optimal number of clusters by selecting the result that is more statistically significant.

**Merging clusters across layers:** Strongly connected clusters from adjacent layers are next merged to uncover multi-layered modules. Consider, $C_l^i, i = 1, 2, ..., K_l$ to be the clusters identified at layer $l$. Let $e_{i,j}^l$ be the number of edges from cluster $C_l^i$ to cluster $C_{l+1}^j$. The two clusters are merged if :

$\frac{e_{i,j}^l}{\sum_{j=1}^{K_{l+1}} e_{i,j}^l} \geq \delta_m$ and $\frac{e_{i,j}^l}{\sum_{i=1}^{K_l} e_{i,j}^l} \geq \delta_m$, where $\delta_m$ is the merging threshold.

The output units are merged with the previous layer's modules, ensuring that $\delta_m$ fraction of incoming edges to the unit are from that module. This allows multiple output units to be matched to the same structural module.

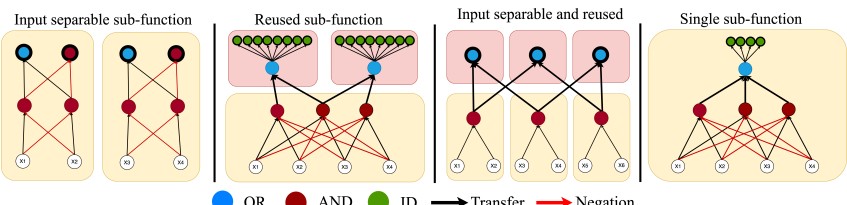

Figure 6: Function graphs used to validate the proposed pipeline

# 5 Experiments and Results

## 5.1 Modular and hierarchical Boolean function graphs

In this section, we conduct experiments on Boolean function graphs with different sub-function properties to validate our pipeline. We begin by testing the pipeline on four function graphs shown in Figure 6. These graphs include: 1) input separable sub-functions, 2) a reused sub-function, 3) sub-functions that are both input separable and reused, and 4) a function graph without any such sub-function where all nodes are strongly connected to every other node.

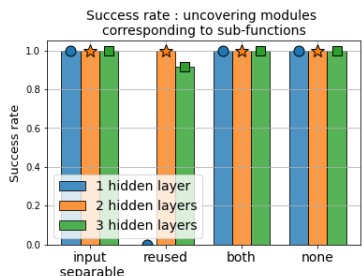

Figure 7: Success rates for the validation function graphs

We perform 36 trials for each function graph by training neural networks with combinations of 3 width values, 3 depth values, and 4 seed values. A trial is considered successful if the proposed pipeline detects a module corresponding to an input separable or reused sub-function (Figure 5). For Boolean functions, we set the modularity metric threshold to -0.2 and the cluster merging threshold to 0.9. Figure 7 shows the success rates for each function graph and NNs with different depths. We observed that the proposed pruning and module detection pipeline has a high success rate when the depth of the NN exceeds that of the function graph. (see to appendix section B for NN visualizations)

## 5.2 Sub-functions that are reused many times are uncovered more accurately

We consider the function graph shown in Figure 6b, which contains a single reused sub-function. We vary the number of times the two intermediate gate nodes in the second hierarchical level are used by decreasing the number of output nodes and measure the success rate. The results are shown in Figure 8, where we observe an increasing trend in the success rate as the number of output nodes using the two gate nodes increases. As the number of output units using the two gate nodes decreases, learning the two gate nodes using the previous "dense" sub-function may not be efficient. We provide visualizations of the corresponding NNs in the appendix section C. Note that NNs with only one hidden layer recover a single module, as the function graphs require four hierarchical levels. If only three hierarchical levels are available, the function graph collapses to a dense graph.

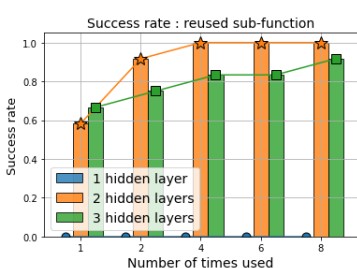

Figure 8: Success rate when the number of times a sub-function is used increases

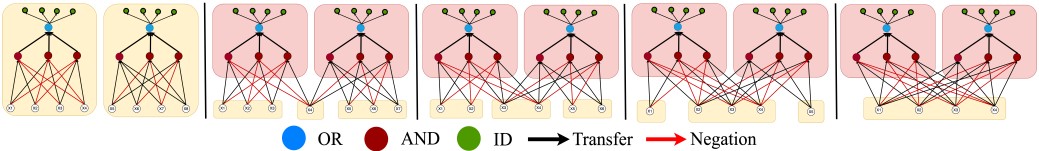

OR ● AND ● ID ● → Transfer → Negation

Figure 9: Increasing the input overlap (reuse) between two input separable sub-functions

## 5.3 Sub-functions with higher input separability are detected more accurately

In this experiment, we consider a function graph with two input separable sub-functions shown in Figure 9a. We increase the overlap between the two separable input sets by decreasing the total number of input nodes and reusing them. We also vary the number of times each sub-function is replicated or used in the output nodes. Our goal is to uncover three properties of the structure: accurately separated input units into sub-function specific and reused, two output sub-functions accurately detected in later layers, and all hidden units belong to either of the two output modules. The success rate for each of these properties as a function of input overlap and sub-function use is shown in Figure 10.

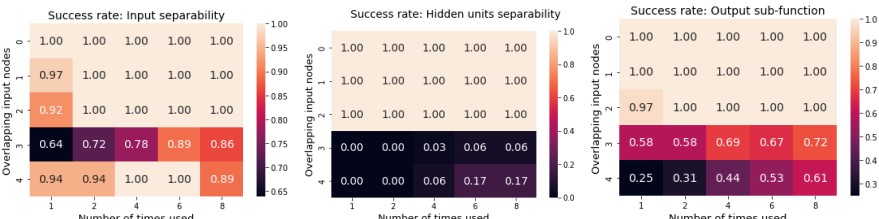

Figure 10: Success rates for various properties uncovered by the pipeline when the input overlap between two sub-functions is increased

We observe that our method has a high success rate for detecting these properties for sub-functions with high input separability. However, as the overlap between input node sets increases, the success rate for detecting input modules decreases. Furthermore, the success rate decreases as the number of times a sub-function is used decreases. We also observe that for sub-functions with low input separability (and high input overlap), intermediate units are often clustered into a single module due to the hidden units pruning step. Finally, we find that the same trend is observed for detecting output sub-functions, where increasing input overlap and decreasing sub-function use results in a low success rate for our method. (See appendix section D for visualization of these results).

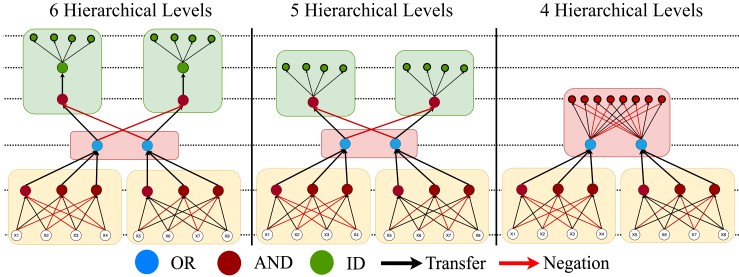

Figure 11: The same Boolean function represented by different function graphs depending on the number of hierarchical levels

## 5.4 Hierarchical structures uncovered vary depending on NN depth

The function graph in Figure 11a consists of two input separable sub-functions, the output of those sub-functions is then used by a single intermediate sub-function. This intermediate sub-function is then reused by two additional sub-functions to produce the final output. Interestingly, we observe that the proposed pipeline uncovers different hierarchical structures depending on the depth of the NN.

Figure 12a shows the success rate of uncovering the specific sub-functions segregated by the NN depth. We find that NNs with depth greater than or equal to the number of hierarchical levels (5) can uncover all three types of sub-functions in the function graph. However, for NNs with lower depth, only the input separable sub-functions are uncovered, while the intermediate and output sub-functions are merged into a single module.

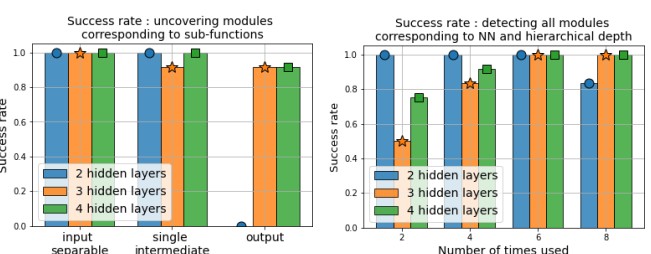

Figure 12: Success rates: a. uncovering specific sub-functions, b. uncovering the overall hierarchical structure, for NNs with varying depths trained on function graph in Figure 11

This result highlights the importance of selecting an appropriate depth for the NN architecture to effectively uncover hierarchical structures in Boolean function graphs. (see appendix section E for NN visualizations) In addition, we report the success rates of uncovering the exact hierarchical structure that corresponds to the depth of the NN as we vary the number of times the output sub-functions

were used (Figure 12b). The findings demonstrate an increasing trend in the success rate, which is consistent with our previous results.

## 5.5 Modular and hierarchical functions with MNIST digits

In this section, we present an experimental evaluation of hierarchically modular tasks constructed using the MNIST handwritten digits dataset. We set the modularity metric threshold to -0.3 and the cluster merging threshold to 0.6 for tasks with MNIST digits.

**Classifying two MNIST digits:** We begin by considering a simple function constructed to classify two MNIST digits. The NN is presented with two images concatenated together, and the output is a vector of length 20. The first 10 indices of the output predict the class for the first image, and the remaining 10 predict the class for the second image. We construct the dataset such that each unique combination of labels has 1000 input data points generated by randomly selecting images. We split the data into training and validation sets with an 8:2 ratio, and then train nine neural networks with varying widths (392, 784, 1568) and number of hidden layers (2, 3, 4) to learn the function for four different seed values. We use the Adam optimizer with bit-wise cross-entropy as the loss function, and select $99\%$ as the accuracy threshold for the pruning algorithm, as all dense NNs achieve at least $99\%$ validation accuracy. Out of the 36 trials conducted, we obtain two completely separable modules for 33 out of 36 trials, indicating the high effectiveness of our approach. (see appendix F for NN visualizations)

**Hierarchical and Modular MNIST Task:** We consider next a hierarchical and modular task that uses the MNIST digits dataset (Figure 13a). The task takes two MNIST images as input, each belonging to the first 8 classes (digits 0-7). The digit values are represented as 3-bit binary vectors and used to construct three output Boolean sub-functions. To generate the dataset, we randomly select 1000 input data points for each unique combination of digits. The data is then split into training and validation sets with an 8:2 ratio. All the NNs that we experiment with reach a validation accuracy of $98\%$, which we use as the accuracy threshold for the pruning algorithm.

To analyze the modular structure uncovered by our methodology, we divide the success rates into three categories: (1) detecting two input separable modules, (2) detecting three output modules, and (3) middle layer unit-separability into either the input separable modules or the output separable modules. We observe that the NNs uncover the three output modules with high success rates. However, for NNs with lower depths, the NNs fail to recover the two input separable modules, and all the units in the early layers are clustered into a single

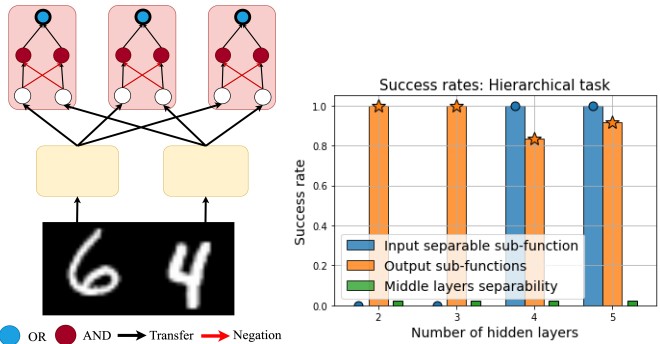

Figure 13: a. Hierarchically modular task constructed with MNIST digits; b. Success rates for various modules recovered by NNs with varying depths.

module. As the depth of the NN increases, we find that the input separable modules are recovered with high accuracy as well. We observe that the success rates for middle layer unit-separability are very low, even when the depth of the NN increases. The units belonging to those hidden layers are often clustered into a single module. These units may be learning representations required to approximate the two digits well. This finding indicates that the NN depths we experiment with may not be sufficient to capture the underlying function, despite the NNs learning the task with high validation accuracy. Please refer to appendix F for NN visualizations.

## 6 Related work

**Emerging modular structures** in trained NNs has been a subject of prior investigation, primarily through unit clustering algorithms. Previous methods for extracting such structures can be categorized

as either *structural* or *functional*. Structural methods organize units based on their structural characteristics, including connectivity and edge-weights [12, 13, 14], while functional methods consider attributes based on unit activation patterns [15, 16]. However, it remains unclear whether these extracted structures represent the underlying hierarchy of sub-functions in a given task.

To the best of our knowledge, this work is the first attempt to introduce an approach that combines training (pruning) and network analysis to unveil the hierarchical modularity of tasks. Our proposed method for clustering units and discovering modules aligns with the structural methods. A series of prior studies leveraged normalized spectral clustering to globally extract unit clusters and analyze their characteristics [14, 15]. Spectral clustering optimizes for N-cuts, quantifying the internal connectivity relative to the external connectivity of unit clusters. Our method is most similar to previously introduced layer-wise unit clustering techniques [12, 13]. These methods consider incoming and outgoing edge-weights to group units within a layer and consolidate edges into single incoming and outgoing connections. Importantly, all previous methods were tailored for conventionally trained NNs. In contrast, our approach is simpler and customized for the pruned NNs we obtain.

**Pruning** of NN units and edges has gained significant traction as a method to enhance NN computational efficiency and reduce memory demands while maintaining performance integrity [21, 24, 23, 34, 35, 30, 36, 37, 38]. Recent research has also focused on investigating how pruning impacts NN generalization [39, 40, 41]. However, prior studies have primarily concentrated on either unit pruning or edge pruning in isolation, without employing both in a sequential manner as proposed in our work. In our approach, we employ unit pruning as a means to condition or compel NNs to learn reused sub-functions effectively. Furthermore, edge pruning is employed to uncover the sparse connectivity between various units or modules.

**Mechanistic interpretability** seeks to reverse-engineer trained NNs to understand their internal mechanisms. One avenue of research within mechanistic interpretability involves identifying the circuits formed within NNs. Recent discoveries include curve-detecting circuits in vision models [42, 43], transformer circuits [44], induction heads [45], and indirect object identification [46], among others. Our proposed approach for uncovering the hierarchical modular task structure dissects the NN into modules responsible for learning specific sub-functions. Hence, simplifying the complexity of reverse engineering entire NNs, by focusing instead on smaller sub-networks.

**Continual learning** aims at learning multiple tasks presented sequentially while preserving performance on previously learned tasks [47, 48, 49, 50]. Modular NNs have been demonstrated to mitigate catastrophic forgetting by freezing and reusing certain modules while introducing and updating others to learn new tasks [6, 7, 8]. However, those methods entail manually designing modular NNs with fixed module sizes at initialization. NNs that naturally acquire a hierarchically modular structure mirroring that of the task may offer additional advantages compared to generic modular NNs. These potential benefits encompass enhanced generalization, efficient transfer learning through the reuse of modules corresponding to frequently utilized sub-functions, and informed module additions to prevent sub-linear increases in NN capacity.

## 7   Conclusion

We have introduced *Neural Sculpting*, a methodology to uncover the hierarchical and modular structure of target tasks in NNs. Our findings first demonstrated that NNs trained conventionally do not naturally acquire the desired structural properties related to input separable and reused sub-functions. To address this limitation, we proposed a training strategy based on iterative pruning of units and edges, resulting in sparse NNs with those previous structural properties. Building upon this, we introduced an unsupervised method to detect modules corresponding to various sub-functions while also uncovering their hierarchical organization. Finally, we validated our proposed methodology by using it to uncover the underlying structure of a diverse set of modular and hierarchical tasks.

As a future research direction, we could investigate the efficiency and theoretical underpinnings of modularity in function graphs, which could further motivate pruning NNs. One potential approach to overcome the computational costs and dependence on initial architecture depth could be to use neural architecture search algorithms to construct modular NNs during training. Additionally, exploring the use of attention mechanisms and transformers to uncover hierarchical modularity in tasks could be an interesting direction for future work. These approaches could provide a more efficient way of obtaining modular NNs that can also better capture the underlying structure of the target task.

## Acknowledgements

This work is supported by the National Science Foundation (Award: 2039741), by the EU's Horizon 2020 Research and Innovation Program (grant agreement no. 739578), and by the Government of the Republic of Cyprus through the Deputy Ministry of Research, Innovation, and Digital Policy. The authors acknowledge the constructive comments given by the NeurIPS 2023 reviewers, Cameron Taylor, Qihang Yao and Mustafa Burak Gurbuz.

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

# A  Appendix structure and details

In this section, we discuss the structure of the appendix. The code and datasets used can be found here: `https://github.com/ShreyasMalakarjunPatil/Neural-Sculpting`.

**Appendix sections (B - F)** accompany section 5 in the main paper and provide visualizations of neural networks (NNs) and the hierarchical modularity uncovered by our proposed pipeline. Each section includes NN visualizations corresponding to the five experiments summarized in the main paper. The main figures in these sections are organized such that the first row represents different function graphs that the NNs are trained to learn, with each column representing a distinct function graph relevant to the experiment. The subsequent rows show visualizations of the uncovered hierarchical modular structures for NNs with increasing depths.

**Visualization figure details:** In the appendix sections, each function graph is associated with visualizations from a specific NN trial at a particular depth value (with one width value and one seed value). This visualization represents the structure that is commonly recovered for the given function graph and NN depth. While the NN width value and seed value remain consistent in the visualizations, they may vary if the recovered structure does not represent the majority. We also present and discuss other structures that are frequently recovered, using other function graphs representing the same Boolean function. For success rate computations, such trials are typically considered failures in recovering overall structures. However, for individual sub-function detection success rates, they may be considered successful if they uncover a module corresponding to that sub-function. Additional visualizations and the hyperparameters used for training and pruning the NNs are detailed in the code base.

**Function graph nomenclature:** In the visualizations, the term "modularity" indicates the function graph being used. The modularity is represented as a list that indicates the number of sub-functions in the hierarchy. For example, if the modularity is $[1, 2]$, it means there are two sub-functions in the upper hierarchical level that take as input the output of a single sub-function lower in the hierarchy. In the code base, we prepend the number of input nodes and append the number of output nodes to this list.

**Appendix section G** accompanies section 2 in the main paper. Section 2 presents two tests developed to demonstrate that NNs, through standard training, do not acquire structural properties reflecting input separable and reused sub-functions. In appendix section G, we provide detailed results for these tests conducted on nine different NN architectures to further support our claims. We also report the results of the same tests conducted on edge-pruned and unit-edge-pruned NNs.

**Appendix section H** accompanies section 3 in the main paper, which presents the method for pruning NNs. This appendix section provides a detailed explanation of the unit and edge pruning algorithms, along with the exploration strategy and results for hyperparameter tuning.

**Appendix section I** accompanies section 4 in the main paper, which introduces the proposed method for detecting modules corresponding to various sub-functions. In this appendix section, we present a detailed algorithm description and the methodology used for hyperparameter exploration.

Note that the pruning and module detection hyperparameters were explored concurrently. However, in the appendix section, we first present experiments with the pruning hyperparameters while keeping the module detection method fixed, and vice versa.

**Resources:** The unit and edge pruning processes for the Boolean functions were executed on a CPU, while the pruning for the MNIST experiment was performed using RTX6000 GPUs.

# B  NN visualizations: Function graphs used for validation

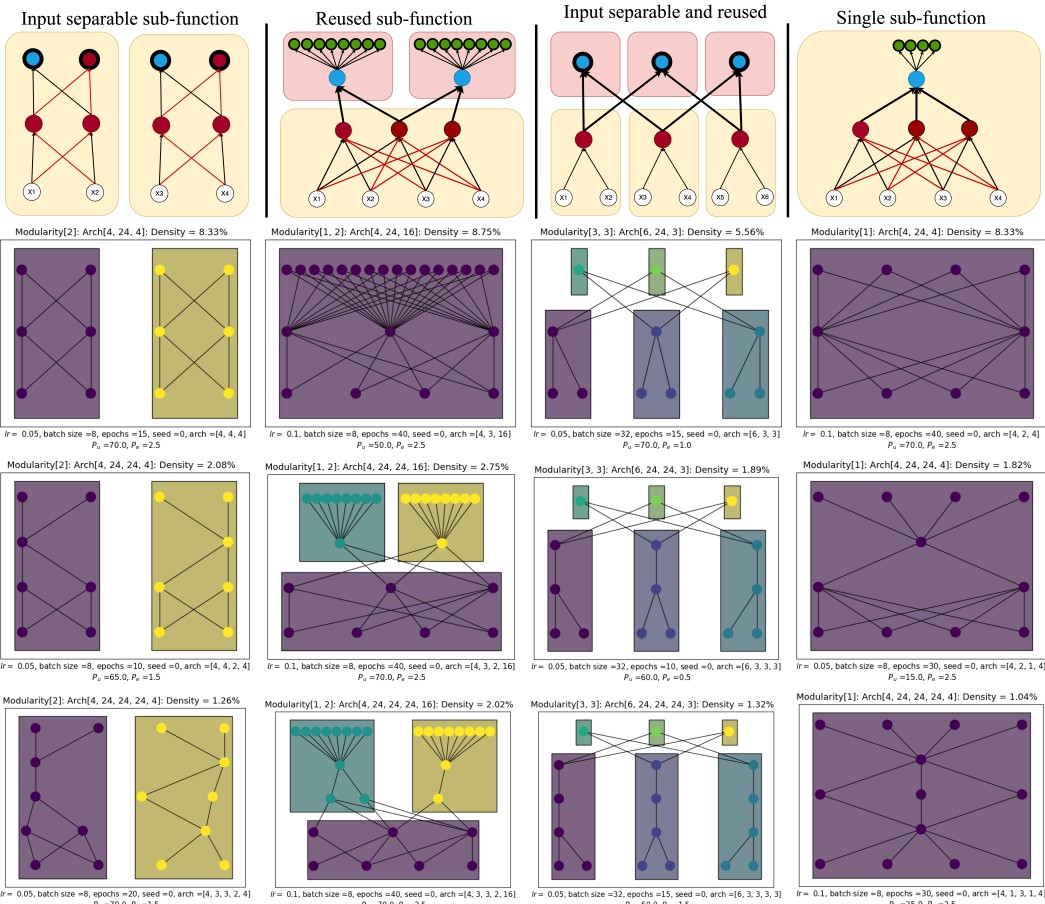

Figure 14: The visualizations provide examples of NNs successfully uncovering the exact hierarchical structure as the validation function graphs. Each column represents a distinct function graph, characterized by: 1. input separable sub-functions, 2. reused sub-function, 3. input separable and reused sub-functions, or 4. a single sub-function. Each row showcases NNs with different depths: 1. one hidden layer, 2. two hidden layers, and 3. three hidden layers.

# C NN visualizations: Function graphs with a single sub-function and increasing reuse

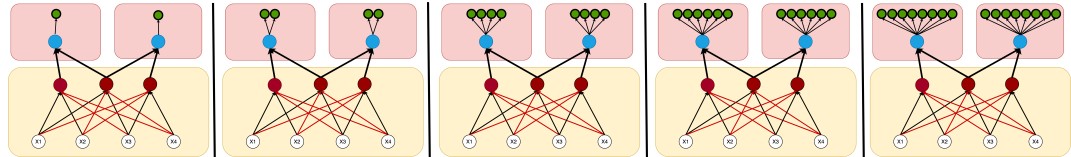

Figure 15: The function graphs consist of a single non-input separable sub-function that is reused to construct two output sub-functions. We vary the number of times the two sub-functions are used or replicated, ranging from 1 to 8.

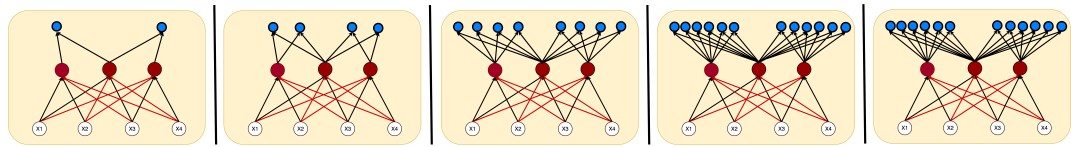

Figure 16: Function graphs in Figure 15 represented using only 3 hierarchical levels.

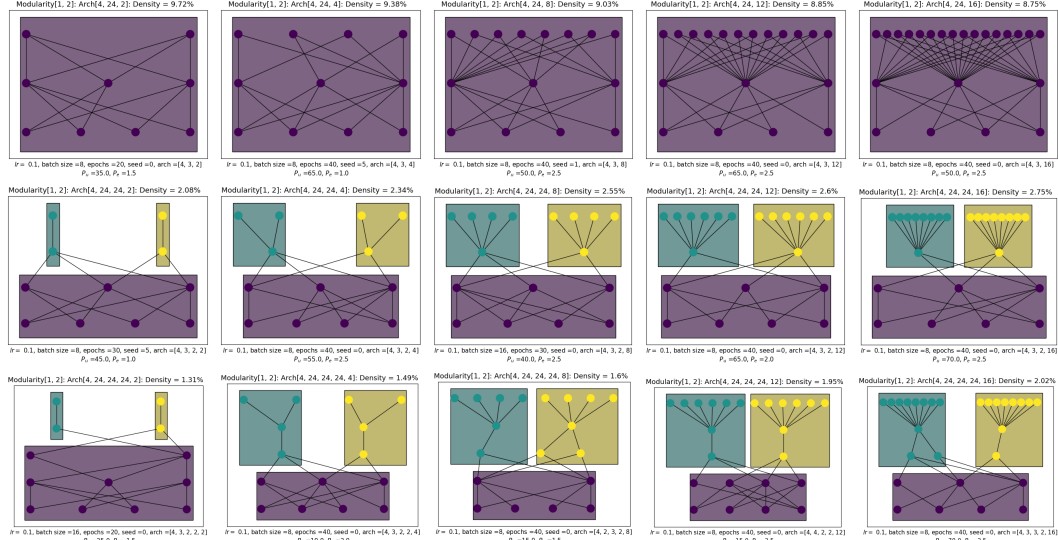

Figure 17: Example visualizations of uncovered modular and hierarchical structure for NNs trained on function graphs in Figure 15. Each row indicates a different NN depth, 1. 1 hidden layer, 2. 2 hidden layers, and 3. 3 hidden layers. Each column indicates a different function graph with sub-function use increasing from 1 to 8. We can observe that when the NN depth is lower than that of the number of hierarchical levels, a single module is recovered. This is consistent with the function graphs that use only 3 hierarchical levels as shown in Figure 16.

# D    NN visualizations : Increasing input overlap and reuse of two input separable sub-functions

## D.1    Completely separable sub-functions

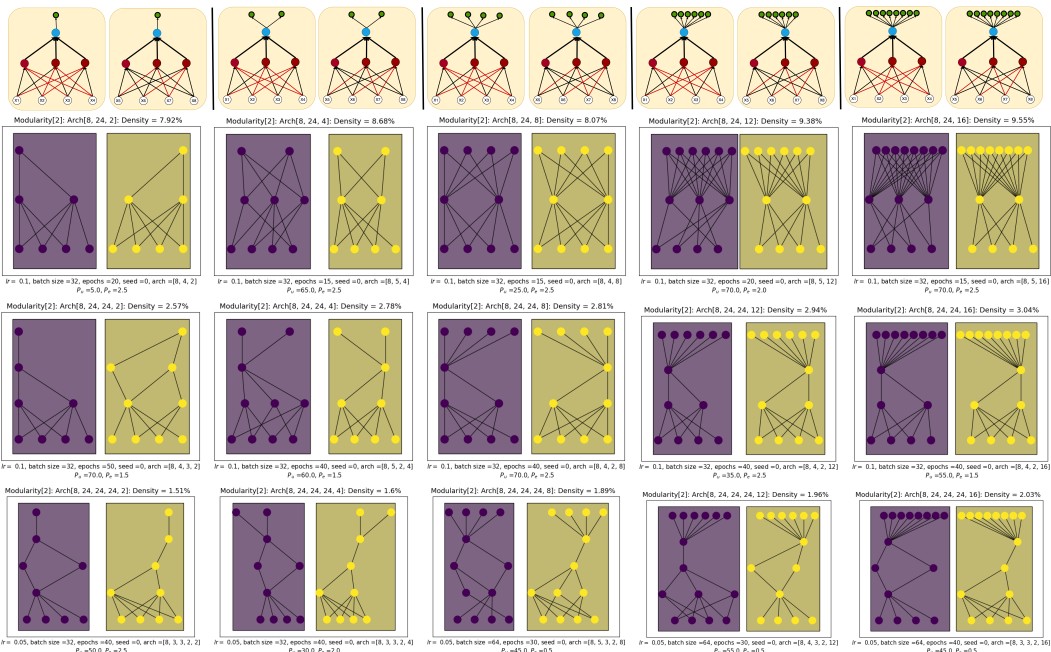

Figure 18: The visualizations presented illustrate the uncovered modular and hierarchical structure of NNs trained on the function graphs depicted in the first row. Within these function graphs, there are two sub-functions that are entirely input separable. Each row in the visualizations corresponds to a different NN depth, namely 1 hidden layer, 2 hidden layers, and 3 hidden layers. Additionally, each column represents a different function graph, where the usage of sub-functions increases incrementally from 1 to 8.

## D.2 Single overlapping input node

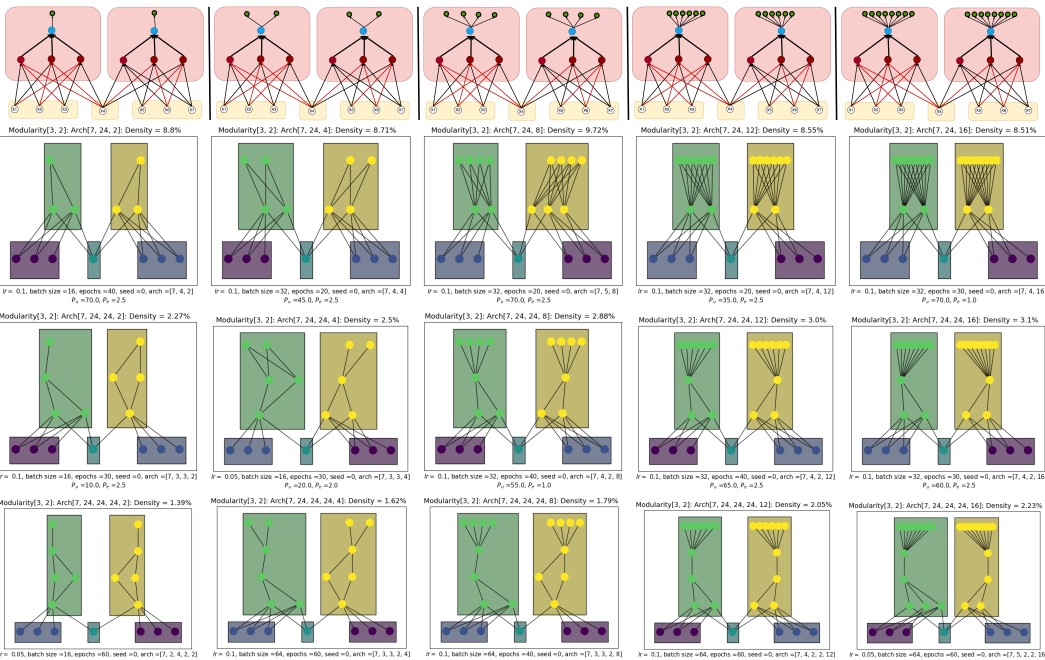

Figure 19: The visualizations provided showcase the uncovered modular and hierarchical structure of NNs trained on function graphs found in the first row. These function graphs comprise two sub-functions that are built using three sets of input nodes. Specifically, there are two sets, each containing three nodes, which are specific to individual sub-functions. The third set consists of one input node that is shared between both sub-functions. In the visualizations, each row corresponds to a different NN depth, ranging from 1 hidden layer to 3 hidden layers. Additionally, each column represents a distinct function graph, with an increasing usage of sub-functions from 1 to 8.

## D.3 Two overlapping input nodes

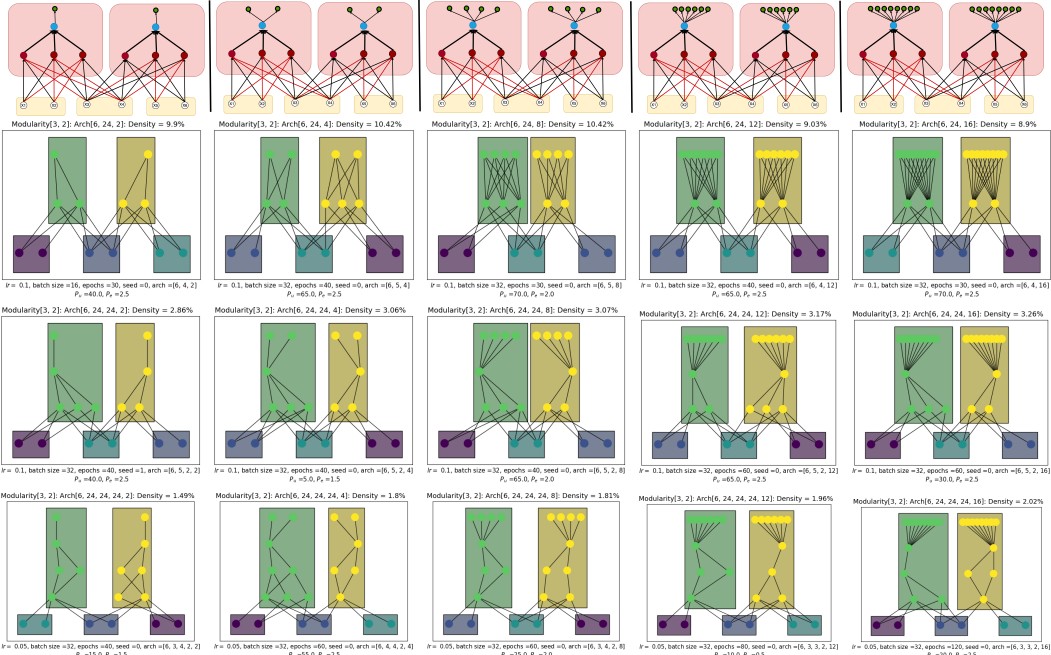

Figure 20: The provided visualizations depict the uncovered modular and hierarchical structure of NNs trained on function graphs found in the first row. These function graphs comprise two sub-functions that are constructed using three sets of input nodes. Within these sets, there are two sub-function-specific sets, each consisting of two nodes, and the third set comprises two input nodes shared between both sub-functions. In the visualizations, each row represents a different NN depth, ranging from 1 hidden layer to 3 hidden layers. Furthermore, each column represents a distinct function graph, with increasing usage of sub-functions from 1 to 8.

## D.4 Three overlapping input nodes

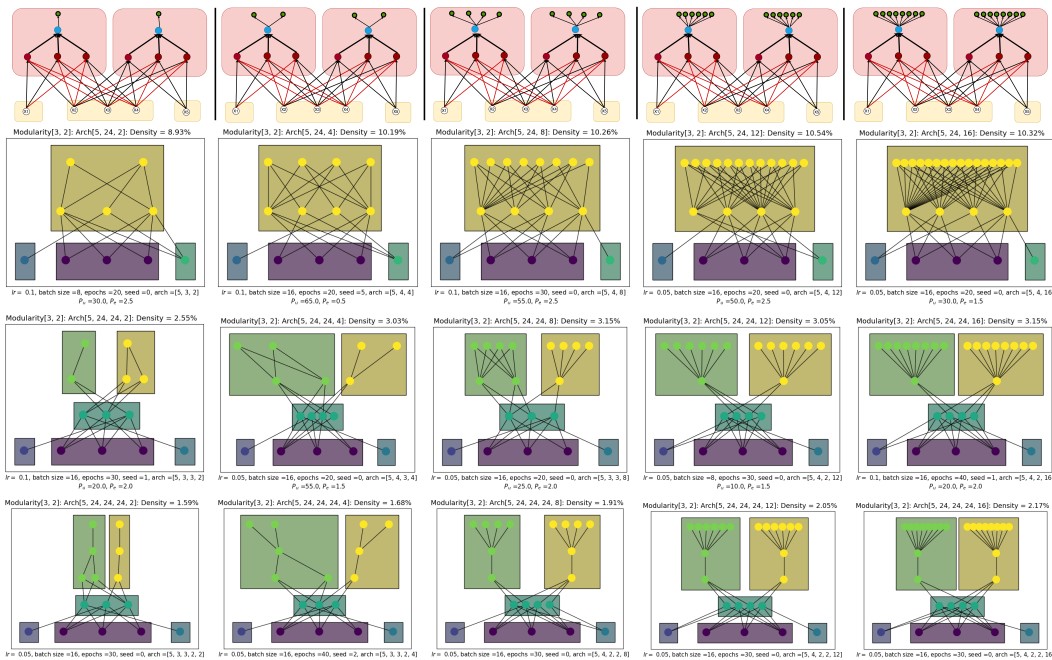

Figure 21: The provided visualizations showcase the most commonly uncovered modular and hierarchical structure for NNs trained on function graphs found in the first row. These function graphs consist of two sub-functions that are constructed using three sets of input nodes. Among these sets, two are sub-function-specific, with each set comprising one node, while the third set consists of three input nodes that are shared between both sub-functions. Each row in the visualizations represents a different NN depth, ranging from 1 hidden layer to 3 hidden layers. Additionally, each column corresponds to a distinct function graph, with an increasing usage of sub-functions from 1 to 8. Observing the visualizations, we can note that the input units are accurately clustered into two modules specific to each sub-function and one reused module. However, the hidden units in the early layers tend to be clustered into a single module more often. This suggests that NNs may learn intermediate states that are reused for both sub-functions when there is a large input reuse set. This phenomenon may be influenced by excessive hidden unit pruning and the existence of learnable hidden states that can be reused independently of input reuse. On the other hand, the output sub-functions are detected accurately for deeper NNs, indicating that the hierarchical organization is better captured as the network depth increases.

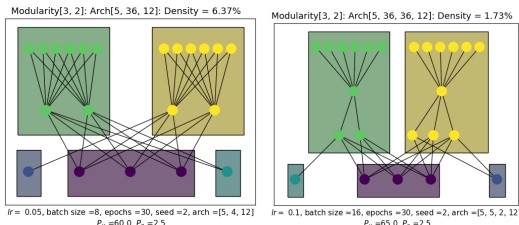

Figure 22: The provided visualizations depict accurately uncovered modular and hierarchical structures for NNs trained on function graphs shown in Figure 21.

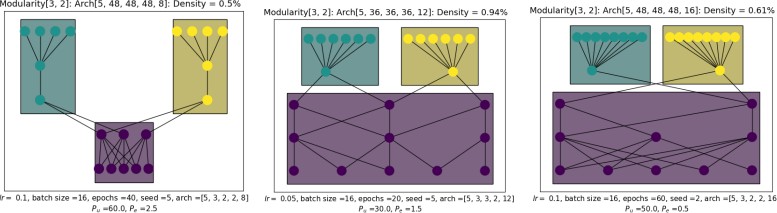

Figure 23: The visualizations highlight the tendency of NNs trained on function graphs from Figure 21 to cluster the input and early hidden layers together while accurately detecting the output sub-functions. This clustering of the input layer and early hidden layers into a single module suggests that the NNs learn representations that are shared among the sub-functions, despite their distinct inputs. This phenomenon becomes more pronounced when the sub-functions are repeatedly utilized within the function graph. It is possible that the network captures common features or patterns that are relevant to all the sub-functions, leading to this shared module in the early layers.

## D.5   All overlapping input nodes

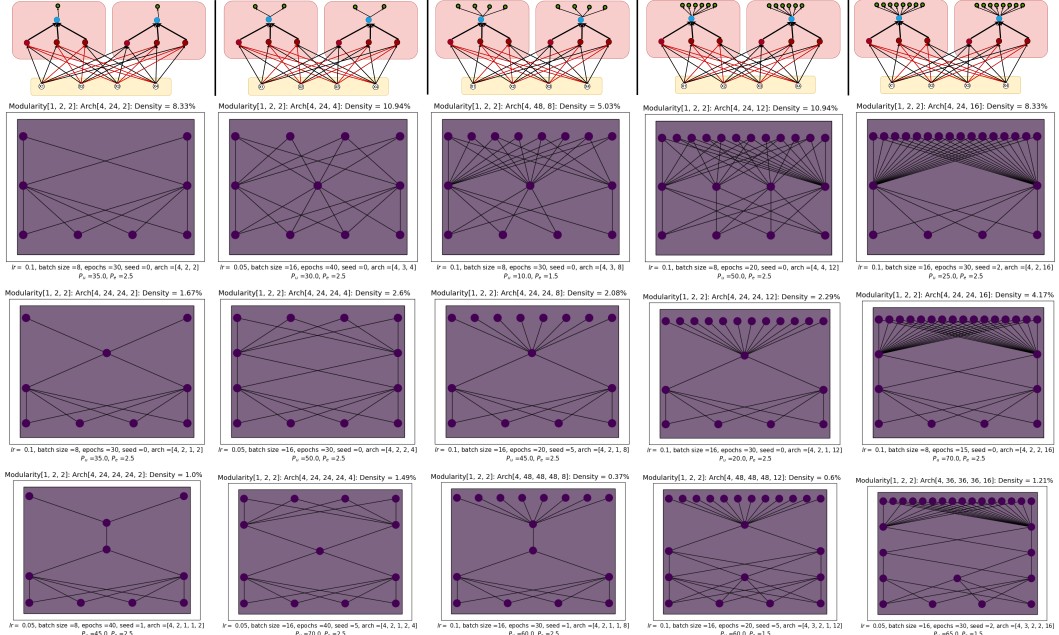

Figure 24: In the set of visualizations for the uncovered modular and hierarchical structure of NNs trained on function graphs in the first row, we can observe a single module being recovered. This finding is consistent with the previous result, where a high degree of input overlap within the function graph, coupled with hidden unit pruning, influences the NN to learn hidden states that are reused for both output sub-functions. Despite the presence of distinct output sub-functions, the network learns to capture common underlying features in all the hidden layers.

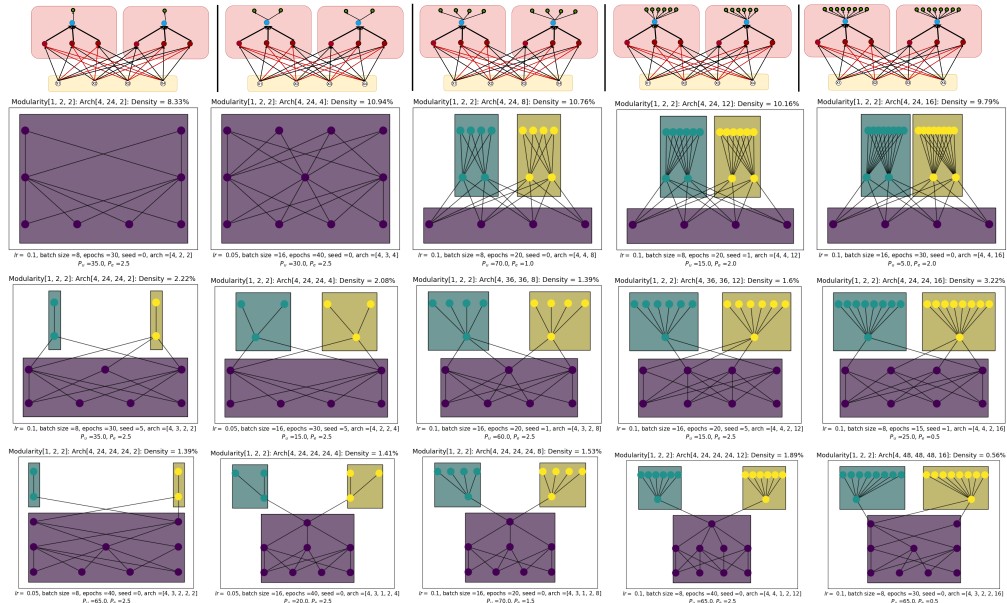

Figure 25: In set 2 of visualizations for the uncovered modular and hierarchical structure of NNs trained on function graphs in the first row, a distinct pattern emerges. In the early layers, the pipeline consistently recovers a single module, primarily because there is no input separability present in the function graphs. However, as we progress to the later layers, we observe a successful separation of units into two distinct modules.

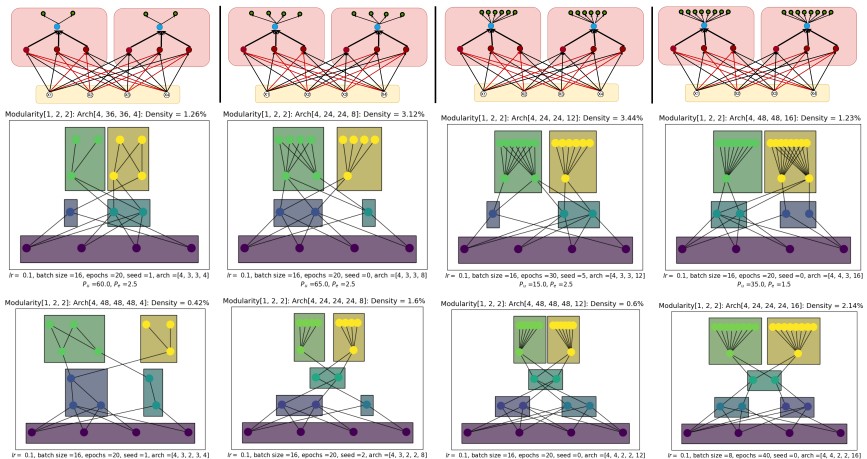

Figure 26: In set 3 of visualizations for the uncovered modular and hierarchical structure of NNs trained on function graphs in the first row, intriguing patterns emerge. The pipeline successfully reveals a single module in the input layer. In later layers, the units are accurately separated into two distinct modules. In NNs with 2 hidden layers, we observe that the first hidden layer has two distinct clusters. One cluster is connected to one of the output modules, while the other cluster is reused by both modules. In NNs with 3 hidden layers, The first hidden layer approximates two sub-functions, which are subsequently reused by both output modules through an intermediate module. We could not interpret those structures using any of the valid function graphs.

# E    NN visualizations : Increasing the number of hierarchical levels

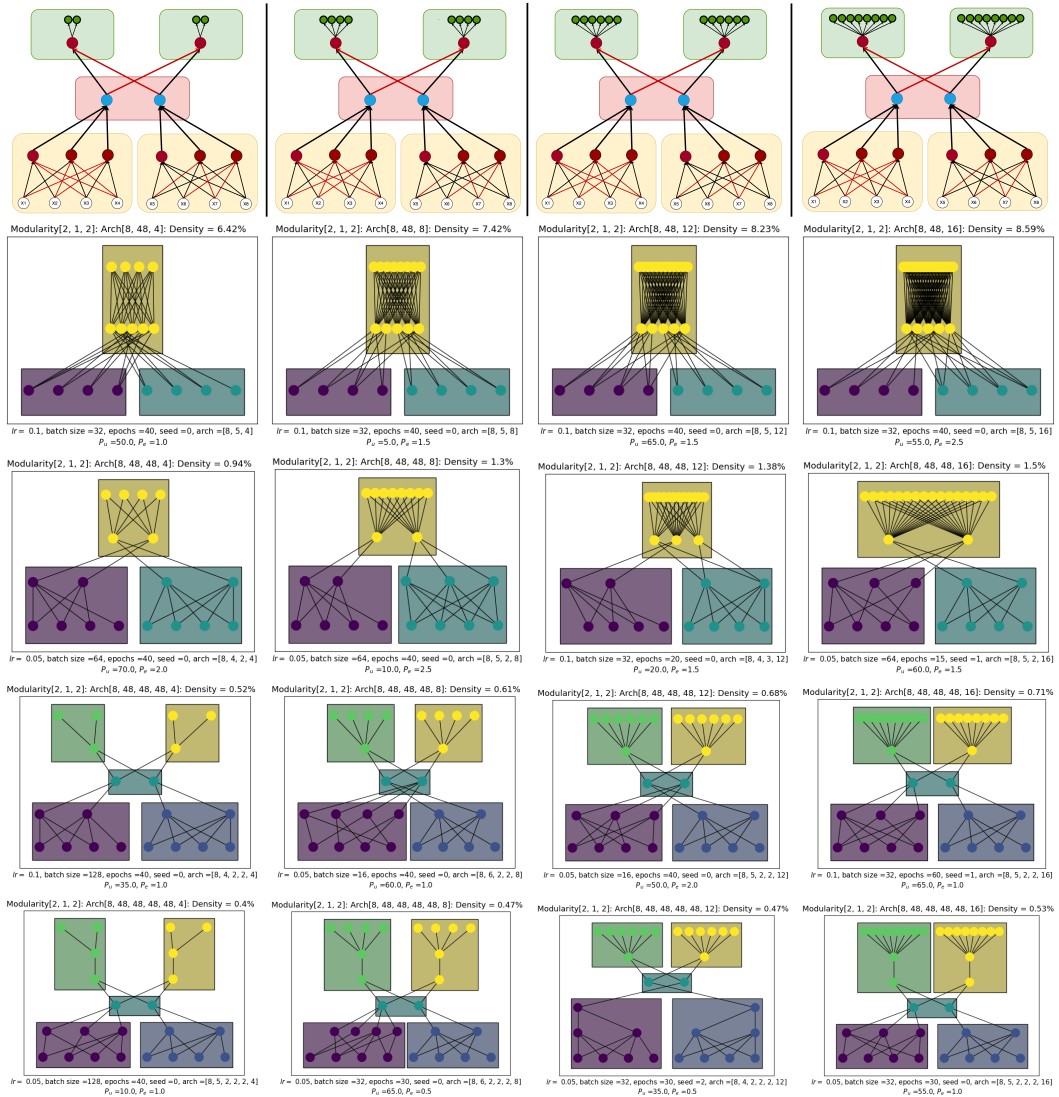

Figure 27: The visualizations of the uncovered modular and hierarchical structure for NNs trained on function graphs in the first row provide valuable insights. Within these function graphs, there are two sub-functions that are completely input separable. The outputs of these sub-functions are then combined into a single sub-function, which is subsequently reused by two output sub-functions. Each row in the visualizations corresponds to a different NN depth, ranging from 1 hidden layer to 4 hidden layers. Additionally, each column represents a different function graph, with an increasing number of sub-function uses ranging from 1 to 8. An interesting observation is that as the depth of the NN varies, the uncovered hierarchical structures also vary. These visualizations shed light on the interplay between network depth and the discovered hierarchical structures, offering insights into how the NNs adapt and represent the relationships among the input separable sub-functions and the output sub-functions.

### E.1 Other structures frequently recovered

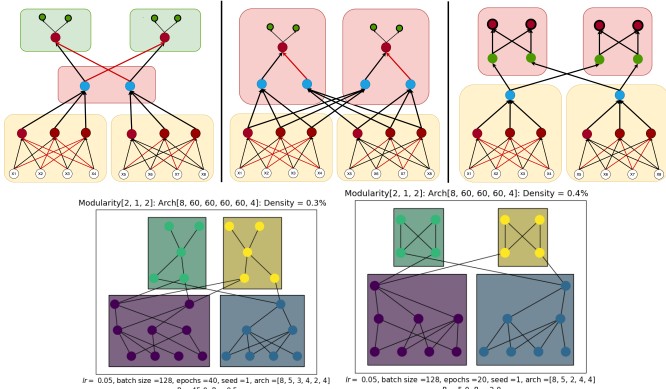

Figure 28: The visualizations showcase other structures that are frequently uncovered by NNs trained on the function graph in the first row. It is evident that the function can also be represented by other function graphs shown in the same row. However, these alternate function graphs are less efficient in terms of the number of edges and gate nodes required. In the first alternate graph, we observe the use of 6 additional edges, while the second alternate graph utilizes 4 additional edges. Moreover, both of these function graphs involve a larger number of gate nodes compared to the original function graph. Many unsuccessful NN trials result in the recovery of these alternate structures, as illustrated in the second row of the visualizations.

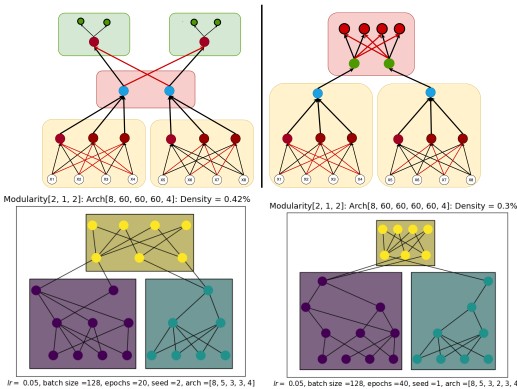

Figure 29: The visualizations showcase other structures that are frequently uncovered by NNs trained on the function graph in the first row. It is evident that the function can also be represented by other function graphs shown in the same row. However, this alternate function graph is less efficient, requiring 2 additional edges compared to the original function graph. Many unsuccessful NN trials result in the recovery of these alternate structures, as illustrated in the second row of the visualizations.

## E.2 Output sub-functions used once

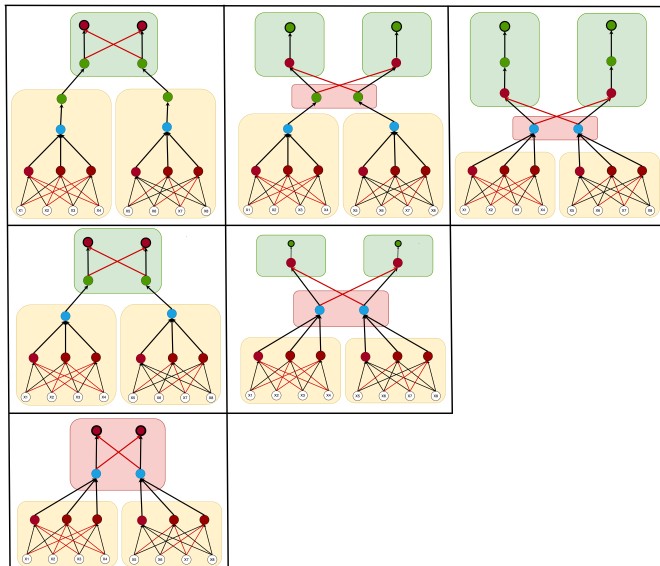

Figure 30: Function graphs representing the Boolean function in 27 when each of the output sub-functions is used only once. In each row of the figure, we can observe function graphs with different numbers of hierarchical levels. Each graph within a row presents an alternate representation of the function, employing the same number of nodes and edges.

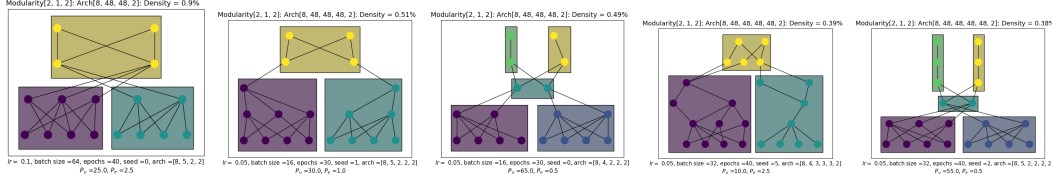

Figure 31: a) Example visualization of the hierarchical structure obtained through an NN with 2 hidden layers, trained on the function illustrated in Figure 30(same structure as shown in Figure 30, row 3a); b) Example visualization of the hierarchical structure obtained through an NN with 3 hidden layers, trained on the function illustrated in Figure 30 (same structure as shown in Figure 30, row 2a); c) Example visualization of the hierarchical structure obtained through an NN with 3 hidden layers, trained on the function illustrated in Figure 30 (same structure as shown in Figure 30, row 2b); d) Example visualization of the hierarchical structure obtained through an NN with 4 hidden layers, trained on the function illustrated in Figure 30 (same structure as shown in Figure 30, row 1a); e) Example visualization of the hierarchical structure obtained through an NN with 4 hidden layers, trained on the function illustrated in Figure 30 (same structure as shown in Figure 30, row 1c).

# F  NN visualizations for tasks with MNIST

## F.1  MNIST task classifying two digits

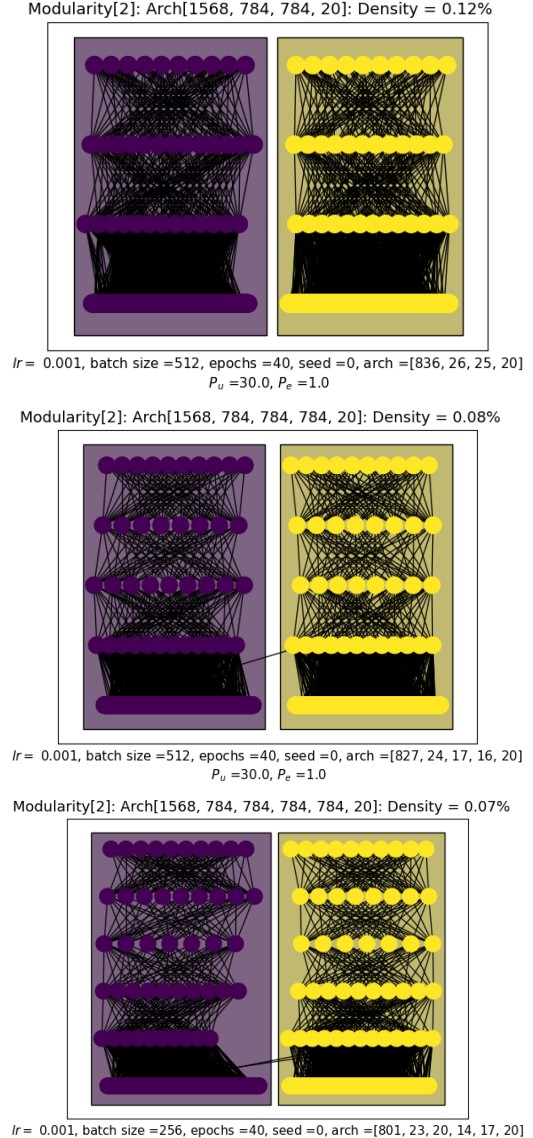

Figure 32: Visualizations of NNs uncovering two modules corresponding to the two MNIST digits classification task with varying depth. Out of the 36 trials conducted, we obtain two completely separable modules for 33 out of 36 trials, indicating the high effectiveness of our approach.

## F.2 Hierarchical task with MNIST digits

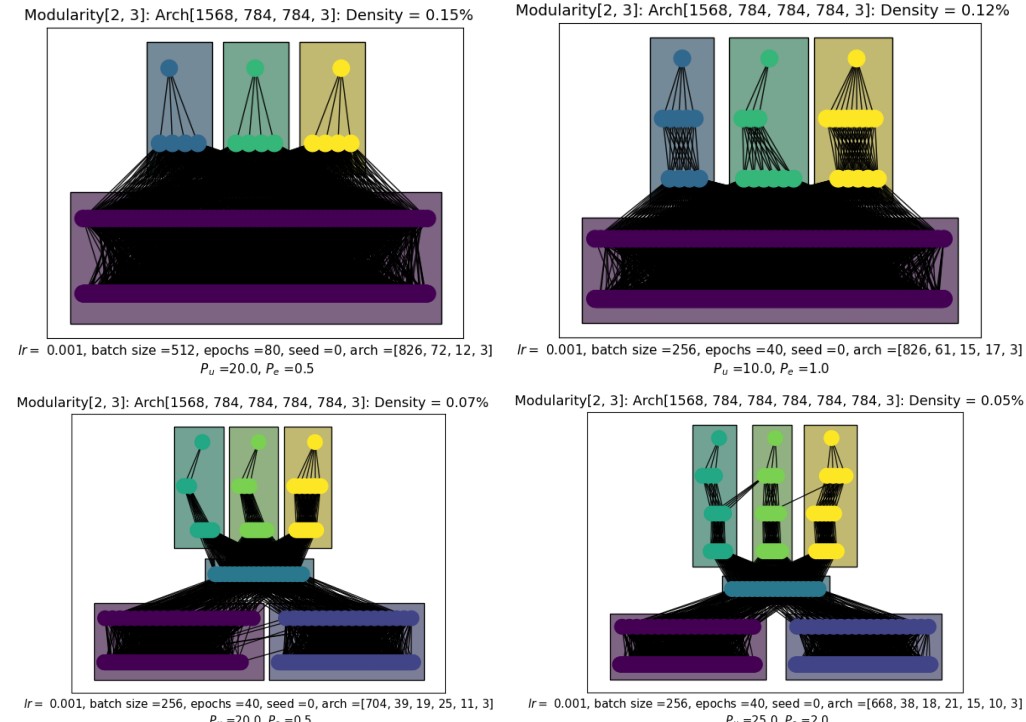

Figure 33: The uncovered NN structures for the hierarchical and modular task with 2 MNIST digit images as input. The NNs uncover the three output modules accurately. The pipeline fails to recover the two input separable modules when the NN depth is low. Deeper NNs do recover two input separable modules. However, the middle layers are clustered into a single module indicating that the depth may be insufficient to detect the two classification modules accurately.

# G   Structural properties of dense and sparse NNs

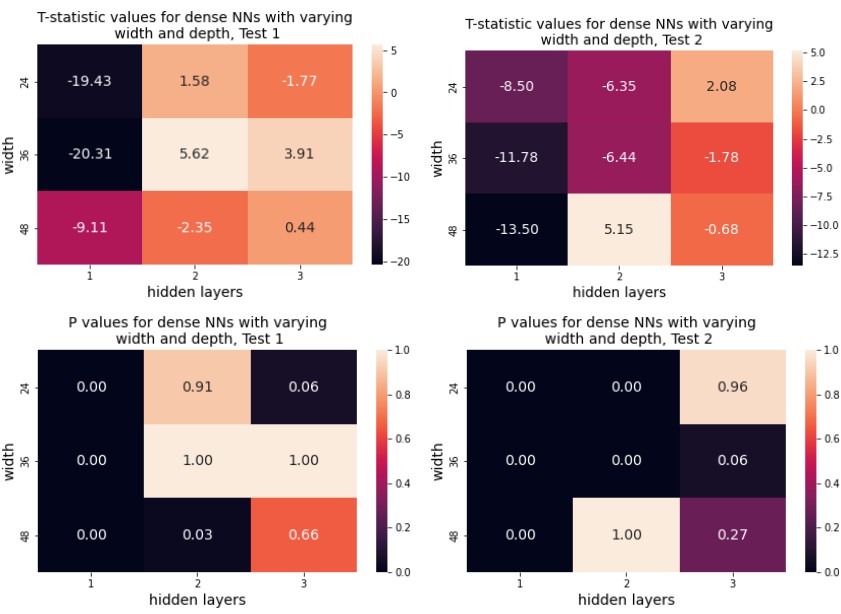

Figure 34: T-statistic and P-values for the two statistical tests conducted to detect input separability in dense NNs. The first row shows the heat maps of t-statistic values (on the two tests) for NNs with varying widths and depths. The second row shows the p-values for the same.

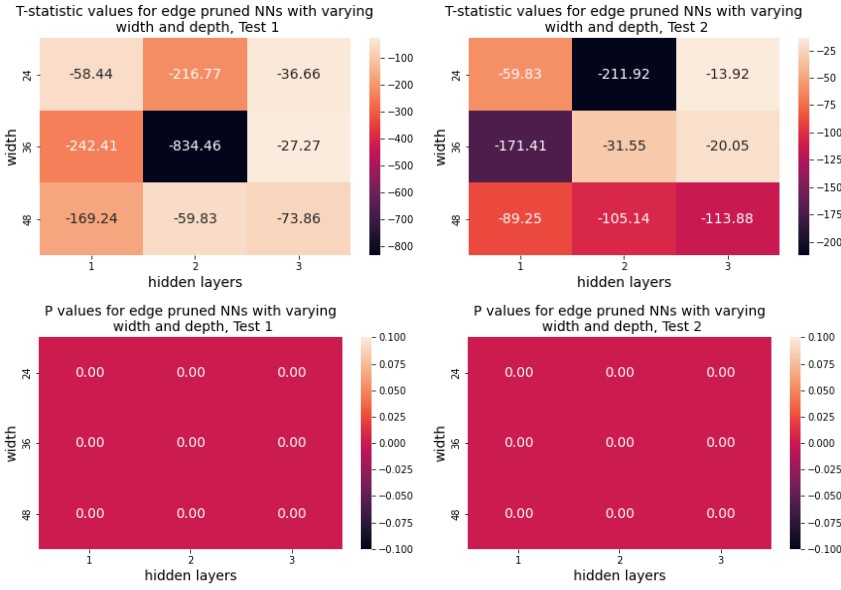

Figure 35: T-statistic and P-values for the two statistical tests conducted to detect input separability in edge-pruned NNs. The first row shows the heat maps of t-statistic values (on the two tests) for NNs with varying widths and depths. The second row shows the p-values for the same.

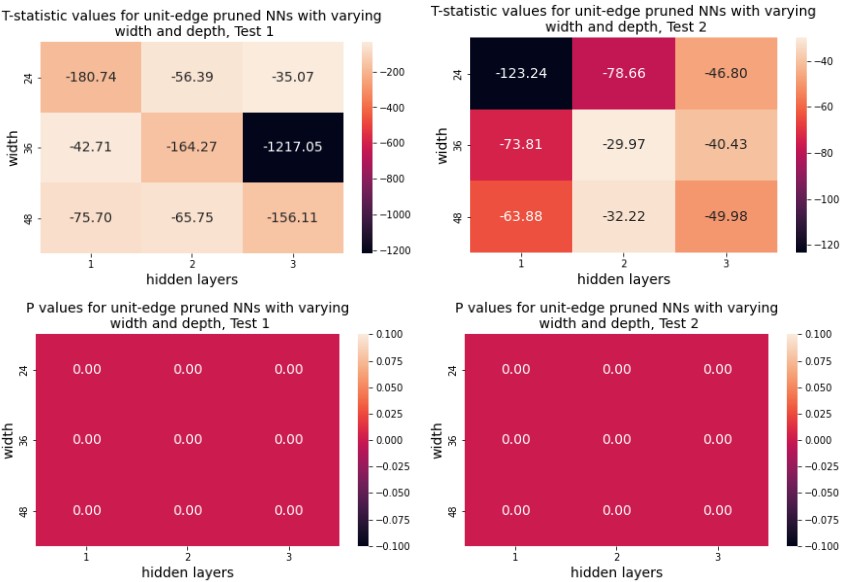

Figure 36: T-statistic and P-values for the two statistical tests conducted to detect input separability in unit and edge pruned NNs. The first row shows the heat maps of t-statistic values (on the two tests) for NNs with varying widths and depths. The second row shows the p-values for the same.

# H Pruning Algorithms

This appendix section is a companion to section 3 of the main paper. Here, we provide a detailed explanation of the pruning algorithm and outline the experiments conducted to explore hyper-parameters for training and pruning NNs.

## H.1 Details

The pseudo code for the unit pruning algorithm can be found in Algorithm 1, while the pseudo code for the edge pruning algorithm is provided in Algorithm 2. In each iteration, pruning is performed (either units or edges), followed by training the NN for the same number of epochs and using the same learning rate schedule.

The pruning procedure consists of two steps. First, a score is assigned to each unit/edge, and then a fraction of the units/edges is pruned based on these scores. It is important to note that the pruning fractions ($p_u$, $p_e$, and $p$) are always computed relative to the original number of hidden units or edges in the NN.

Both unit pruning and edge pruning are implemented using binary masks. Each edge weight is assigned a binary mask value, where a value of 1 indicates that the edge weight is not pruned, and a value of 0 indicates that it is pruned. When a unit is pruned, all incoming and outgoing edge weights associated with that unit are masked out by setting their binary mask values to 0.

**Unit and edge pruning gradients:** In our experiments, we conducted a grid search over the $p_u$ and $p_e$ values for each NN architecture, function graph, and seed value. We explored a range of 14 different $p_u$ values, ranging from $5\%$ to $70\%$ in increments of $5\%$, and 5 different $p_e$ values: $0.5, 1.0, 1.5, 2.0, 2.5$. It is important to note that the $p_u$ values used throughout the process are relative to the original number of units in the NN, while the $p_e$ values are relative to the number of edges in the original NN. After the grid search, we select the NN with the lowest density among all the sparse NNs obtained. Typically, we find that higher unit pruning gradients ($p_u > 40$) work best, as they allow for the removal of a lower number of units in each iteration by halving the gradient value.

---

**Algorithm 1:** Unit pruning algorithm

---

**Data:** $\boldsymbol{\theta} \in \mathbb{R}^a, (\boldsymbol{\mathcal{X}}_t, \boldsymbol{\mathcal{Y}}_t), (\boldsymbol{\mathcal{X}}_v, \boldsymbol{\mathcal{Y}}_v), p_u$, scoring metric, $f_s(\boldsymbol{\theta}, (\boldsymbol{\mathcal{X}}_v, \boldsymbol{\mathcal{Y}}_v))$ (loss sensitivity)
**Result:** mask, $\boldsymbol{M}$
$\boldsymbol{\theta} \leftarrow train(\boldsymbol{\theta}, (\boldsymbol{\mathcal{X}}_t, \boldsymbol{\mathcal{Y}}_t))$ ;                            /* Train the dense NN */
$A_v \leftarrow Acc(\boldsymbol{\theta}, (\boldsymbol{\mathcal{X}}_v, \boldsymbol{\mathcal{Y}}_v))$;
$p \leftarrow 0$ ;                        /* Assign initial pruning percentage as 0 */
$p_{min} \leftarrow 100/(\sum N_l)$ ;            /* minimum pruning percentage difference */
$prune \leftarrow True$;
**while** *prune == True* **do**
  $p = p + p_u$;
  $S \leftarrow f_s(\boldsymbol{\theta}, (\boldsymbol{\mathcal{X}}_v, \boldsymbol{\mathcal{Y}}_v))$ ;   /* Pruning scores equal to the loss sensitivity */
  $T \leftarrow percentile(S, p)$;
  $\boldsymbol{M}_p \leftarrow \mathbb{I}(S > T)$ ;                    /* Threshold the scores to obtain mask */
  $\boldsymbol{\theta}_p \leftarrow \boldsymbol{\theta} \odot \boldsymbol{M}_p$;
  $\boldsymbol{\theta}_p \leftarrow train(\boldsymbol{\theta}_p, (\boldsymbol{\mathcal{X}}_t, \boldsymbol{\mathcal{Y}}_t))$ ;                            /* Train the NN */
  $A \leftarrow acc(\boldsymbol{\theta}_p, (\boldsymbol{\mathcal{X}}_v, \boldsymbol{\mathcal{Y}}_v))$;
  **if** $A >= A_v$ **then**
  $\quad | \quad \boldsymbol{\theta} \leftarrow \boldsymbol{\theta}_p$;
  **else**
  $\quad | \quad p \leftarrow p - p_u$;
  $\quad | \quad p_u \leftarrow p_u/2$;
  $\quad | \quad$ **if** $p_u < p_{min}$ **then**
  $\quad | \quad | \quad prune \leftarrow False$;
  $\quad | \quad$ **end**
  **end**
**end**

---

**Algorithm 2:** Edge pruning algorithm

---

**Data:** $\boldsymbol{\theta} \in \mathbb{R}^a, (\boldsymbol{\mathcal{X}}_t, \boldsymbol{\mathcal{Y}}_t), (\boldsymbol{\mathcal{X}}_v, \boldsymbol{\mathcal{Y}}_v), p_e$
**Result:** mask, $\boldsymbol{M}$
$\boldsymbol{\theta} \leftarrow train(\boldsymbol{\theta}, (\boldsymbol{\mathcal{X}}_t, \boldsymbol{\mathcal{Y}}_t))$ ;                            /* Train the dense NN */
$A_v \leftarrow Acc(\boldsymbol{\theta}, (\boldsymbol{\mathcal{X}}_v, \boldsymbol{\mathcal{Y}}_v))$;
$p \leftarrow 0$ ;                        /* Assign initial pruning percentage as 0 */
$p_{min} \leftarrow 100/a$ ;              /* minimum pruning percentage difference */
$prune \leftarrow True$;
**while** *prune == True* **do**
  $p = p + p_e$;
  $S \leftarrow |\boldsymbol{\theta}|$ ;          /* Pruning scores equal to the weight magnitude */
  $T \leftarrow Percentile(S, p)$;
  $\boldsymbol{M}_p \leftarrow \mathbb{I}(S > T)$ ;                 /* Threshold the scores to obtain mask */
  $\boldsymbol{\theta}_p \leftarrow \boldsymbol{\theta} \odot \boldsymbol{M}_p$;
  $\boldsymbol{\theta}_p \leftarrow train(\boldsymbol{\theta}_p, (\boldsymbol{\mathcal{X}}_t, \boldsymbol{\mathcal{Y}}_t))$ ;                            /* Train the NN */
  $A \leftarrow Acc(\boldsymbol{\theta}_p, (\boldsymbol{\mathcal{X}}_v, \boldsymbol{\mathcal{Y}}_v))$;
  **if** $A >= A_v$ **then**
  $\quad | \quad \boldsymbol{\theta} \leftarrow \boldsymbol{\theta}_p$
  **else**
  $\quad | \quad p \leftarrow p - p_e$;
  $\quad | \quad p_e \leftarrow p_e/2$;
  $\quad | \quad$ **if** $p_e < p_{min}$ **then**
  $\quad | \quad | \quad prune \leftarrow False$;
  $\quad | \quad$ **end**
  **end**
**end**

---

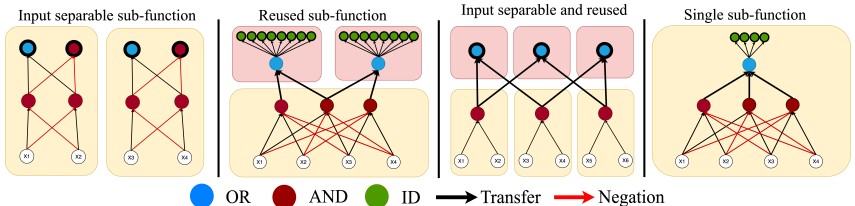

Figure 37: Function graphs used to validate the proposed pipeline

Table 2: Success rates for NNs with varying learning rate and batch size, used for training on function graph with a. input separable sub-functions, and b. a reused sub-function.

| Input separable sub-functions | | | | | Reused sub-function | | | | |
|---|---|---|---|---|---|---|---|---|---|
| lr | 0.005 | 0.01 | 0.05 | 0.1 | lr | 0.005 | 0.01 | 0.05 | 0.1 |
| batch size | | | | | batch size | Reused | | | |
| NNs with 1 Hidden Layer | | | | | NNs with 1 Hidden Layer | | | | |
| 1 | 1.0 | ✗ | 1.0 | ✗ | 1 | 0.5 | 0.5 | 1.0 | 1.0 |
| 2 | 1.0 | 1.0 | 1.0 | 1.0 | 2 | 0.5 | 0.5 | 1.0 | 1.0 |
| 4 | 1.0 | 1.0 | 1.0 | 1.0 | 4 | 0.5 | 1.0 | 1.0 | 1.0 |
| 8 | 1.0 | 1.0 | 1.0 | 1.0 | 8 | ✗ | 0.5 | 0.75 | 1.0 |
| 16 | ✗ | 1.0 | 1.0 | 1.0 | 16 | ✗ | ✗ | 1.0 | 1.0 |
| NNs with 2 Hidden Layers | | | | | NNs with 2 Hidden Layers | | | | |
| 1 | 1.0 | 1.0 | 1.0 | ✗ | 1 | 0.25 | 0.25 | 0.5 | ✗ |
| 2 | 1.0 | 1.0 | 1.0 | ✗ | 2 | 0.5 | 0.0 | 0.75 | 1.0 |
| 4 | 1.0 | 1.0 | 1.0 | 1.0 | 4 | 0.0 | 0.25 | 1.0 | 1.0 |
| 8 | 1.0 | 1.0 | 1.0 | 1.0 | 8 | 0.25 | 0.0 | 0.5 | 1.0 |
| 16 | 1.0 | 1.0 | 1.0 | 1.0 | 16 | ✗ | 0.25 | 0.75 | 0.5 |
| NNs with 3 Hidden Layers | | | | | NNs with 3 Hidden Layers | | | | |
| 1 | 1.0 | 1.0 | ✗ | ✗ | 1 | 0.25 | 0.5 | ✗ | ✗ |
| 2 | 1.0 | 1.0 | ✗ | ✗ | 2 | 0.25 | 0.5 | ✗ | ✗ |
| 4 | 1.0 | 1.0 | 1.0 | ✗ | 4 | 0.5 | 0.5 | 0.75 | ✗ |
| 8 | 1.0 | 1.0 | 1.0 | ✗ | 8 | 0.25 | 0.25 | 0.75 | 1.0 |
| 16 | 1.0 | 1.0 | 1.0 | 1.0 | 16 | ✗ | ✗ | 0.75 | 1.0 |

## H.2 Learning rate and batch size experiments

In this section, we conduct experiments to investigate the impact of learning rates and batch sizes on the structures obtained through pruning. Previous studies have examined the roles of these hyper-parameters in terms of convergence, generalization, and the ability of NNs to find global optima [51, 52]. However, most of these studies train NNs for a fixed number of epochs or iterations, which may not be suitable for our experiment as we require the original dense NNs to learn the task effectively.

To determine the appropriate number of epochs, we aim for the NNs with the four considered seed values to achieve $100\%$ validation accuracy. We set a maximum of 120 epochs to avoid excessively long training or pruning times. Consequently, some combinations of learning rates and batch sizes do not achieve $100\%$ validation accuracy. These combinations typically fall into two categories: 1) high learning rates and low batch sizes leading to unstable training, and 2) low learning rates and high batch sizes resulting in an insufficient number of training iterations to achieve $100\%$ validation accuracy. Such cases are marked with an ✗ in the result tables.

We validate and tune hyperparameters for our pipeline using four function graphs shown in Figure 37. These graphs include input separable sub-functions, reused sub-functions, a combination of input separable and reused sub-functions, and a non-modular function graph. We then train and prune NNs with a width of 24 and varying depths, reporting the success rates for detecting the same hierarchical structure as the function graph.

Table 3: Success rates for NNs with varying learning rate and batch size, used for training on function graph with a. input separable and reused sub-functions, and b. a single sub-function (non-modular graph).

**a.**

| lr | 0.005 | 0.01 | 0.05 | 0.1 |
|---|---|---|---|---|
| batch size | | | | |
| NNs with 1 Hidden Layer | | | | |
| 1 | 0.75 | 1.0 | 1.0 | ✗ |
| 2 | 0.75 | 0.75 | 1.0 | 1.0 |
| 4 | 0.75 | 0.5 | 1.0 | 1.0 |
| 8 | 0.5 | 0.75 | 1.0 | 1.0 |
| 16 | 0.75 | 1.0 | 1.0 | 1.0 |
| 32 | 0.75 | 1.0 | 1.0 | 1.0 |
| 64 | ✗ | 0.75 | 1.0 | 1.0 |
| NNs with 2 Hidden Layers | | | | |
| 1 | 0.75 | 1.0 | 0.25 | ✗ |
| 2 | 1.0 | 1.0 | 0.25 | ✗ |
| 4 | 0.75 | 1.0 | 0.0 | ✗ |
| 8 | 1.0 | 1.0 | 1.0 | ✗ |
| 16 | 1.0 | 1.0 | 1.0 | 0.25 |
| 32 | 1.0 | 1.0 | 1.0 | 0.5 |
| 64 | 1.0 | 0.75 | 1.0 | 1.0 |
| NNs with 3 Hidden Layers | | | | |
| 1 | 1.0 | 1.0 | ✗ | ✗ |
| 2 | 0.75 | 1.0 | ✗ | ✗ |
| 4 | 1.0 | 0.75 | 0.5 | ✗ |
| 8 | 0.75 | 1.0 | 0.75 | 0.25 |
| 16 | 0.5 | 0.75 | 0.5 | 0.5 |
| 32 | 0.5 | 0.75 | 1.0 | 0.5 |
| 64 | 0.5 | 0.75 | 1.0 | 0.5 |

**b.**

| lr | 0.005 | 0.01 | 0.05 | 0.1 |
|---|---|---|---|---|
| batch size | Single | | | |
| 1 | 1.0 | 1.0 | 1.0 | 1.0 |
| 2 | 1.0 | 1.0 | 1.0 | 1.0 |
| 4 | 1.0 | 1.0 | 1.0 | 1.0 |
| 8 | ✗ | 1.0 | 1.0 | 1.0 |
| 16 | ✗ | ✗ | 1.0 | 1.0 |
| NNs with 2 Hidden Layers | | | | |
| 1 | 1.0 | 1.0 | 1.0 | ✗ |
| 2 | 0.75 | 0.75 | 1.0 | 1.0 |
| 4 | 1.0 | 0.75 | 1.0 | 1.0 |
| 8 | 0.75 | 1.0 | 1.0 | 1.0 |
| 16 | 1.0 | 1.0 | 1.0 | 1.0 |
| NNs with 3 Hidden Layers | | | | |
| 1 | 0.25 | 1.0 | 1.0 | ✗ |
| 2 | 0.5 | 0.75 | 0.5 | ✗ |
| 4 | 0.75 | 1.0 | 1.0 | ✗ |
| 8 | 1.0 | 1.0 | 1.0 | 1.0 |
| 16 | 1.0 | 0.75 | 0.75 | 1.0 |

Tables 2 and 3 present the success rates for the four different function graphs considered. We observe that higher learning rates generally lead to more successful trials across all function graphs and NN depths. This aligns with previous findings indicating that higher learning rates result in more globally optimal solutions [53, 54]. On the other hand, lower learning rates tend to under-prune the NNs, resulting in inefficient hierarchical structures in terms of NN density.

When using high learning rates and low batch sizes, we find that training becomes highly unstable, often leading to NNs failing to achieve 100% validation accuracy for all seed values. Therefore, when training and pruning the NNs for other function graphs, we explore the training regime of high learning rates combined with high batch sizes.

# I   Unit clustering algorithm

## I.1   Clustering algorithm hyper-parameter experiments

In this section, we discuss the hyperparameters used in the proposed unit clustering algorithm. These hyperparameters include the distance measure for clustering units, the linkage method in Agglomerative clustering, and the criteria for selecting the optimal number of clusters. The primary criterion for the algorithm's usefulness is its ability to uncover modules and a hierarchical structure that resembles the hierarchical modularity of the function when using a fixed set of hyperparameters.

We consider the four function graphs mentioned in the main part of the paper, which include graphs with input separable sub-functions, a reused sub-function, both input separable and reused sub-functions, and a non-modular function graph. We train and prune a wide range of NN architectures for each function graph and pass the resulting sparse NNs through all combinations of the aforementioned

**Algorithm 3:** Unit clustering algorithm for a given layer $l$

---

**Data:** $f_i^l, i = 1, ..., N_l, t_m$ (modularity metric threshold) ; /* Given the feature vectors */

**Result:** Optimal number of clusters, $K_l$

$k \leftarrow 2$ ;         /* Enumerate through different cluster assignments */

**while** $k < N_l$ **do**

     $C_k \leftarrow Agglo(f_l, k)$ ;     /* Agglomerative clustering dendogram cut giving $k$ clusters */

     $s_k \leftarrow Metric(C_k)$ ;    /* Compute the metric to asses clustering quality */

     **if** $s_k \leq s$ **then**

         $K_l \leftarrow k$ ;        /* Find the number of clusters with lowest score */

         $s \leftarrow s_k$;

     **end**

     $k \leftarrow k + 1$;

**end**

$Z_{sin} \leftarrow 0$;

$Z_{sep} \leftarrow 0$;

**if** $K_l == 2$ *or* $K_l == N_l - 1$ *or* $s > t_m$ **then**

     $Z_{sin} \leftarrow$ test 2 ;          /* Run test for single cluster detection */

     $Z_{sep} \leftarrow$ test 1 ;          /* Run test for separate clusters detection */

**end**

**if** $Z_{sin} > 0$ *or* $Z_{sep} > 0$ **then**

     **if** $Z_{sin} \geq Z_{sep}$ **then**

         $K_l \leftarrow 1$ ;        /* If either one of the test result is positive */

     **else**

         $K_l \leftarrow N_l$;

     **end**

**end**

---

hyperparameters. We then select the set of hyperparameters that successfully uncover the modular structure for the maximum number of NNs and function graphs.

We employ the Agglomerative clustering method on the feature vectors, initially assigning each unit to its own cluster. In each iteration, two clusters are merged into one, and this process continues until all clusters are merged into a single cluster. The specific method used to select the two clusters to merge in each iteration is known as the linkage method. We experiment with four types of linkage methods: average, single, complete, and ward. Average linkage selects the two clusters to merge based on the minimum average pairwise distance between unit features in the two clusters, while single uses the minimum pairwise distance and complete uses the maximum pairwise distance. Ward linkage selects the two clusters to merge based on the cluster variance of the newly formed cluster.

Agglomerative clustering produces multiple cluster assignments depending on the number of clusters. Each iteration of the clustering method yields a cluster assignment with a different number of clusters, ranging from $k = 1$ to $N$, where $N$ is the number of units. To determine the optimal number of clusters, we experiment with scoring metrics and select the $k$ value with the best cluster quality at each iteration. Specifically, we explore four different metrics: 1. Silhouette Index [55], 2. CH-index [56], 3. DB-index [57], and 4. Modularity metric (modified).

From Table 4, we observe that all clustering linkage methods combined with either the modularity metric or Silhouette index perform well. Therefore, we choose the combination of average linkage, cosine distances, and the modularity metric. We discard Euclidean distance as the feature vectors are binary, and we select cosine distance over Jaccard distance to ensure appropriate clustering even in the presence of noise in the feature vectors. Additionally, we prefer average linkage over complete and single linkage methods to enhance the clustering method's robustness to noise.

Table 4: Success rates for different clustering algorithm hyper-parameters on 4 function graphs and NNs with varying widths and depths (Figure 37)

| Clustering Linkage | Distance Measure | Modularity metric | Silhouette index | CH-Index | DB-Index |
|---|---|---|---|---|---|
| Feature vector step size $s = L$ | | | | | |
| Complete | Jaccard | 0.99 | 0.99 | 0.49 | 0.49 |
| Single | Jaccard | 0.99 | 0.99 | 0.49 | 0.49 |
| Average | Jaccard | 0.99 | 0.99 | 0.49 | 0.49 |
| Complete | Cosine | 0.99 | 0.99 | 0.49 | 0.49 |
| Single | Cosine | 0.99 | 0.99 | 0.49 | 0.49 |
| Average | Cosine | 0.99 | 0.99 | 0.49 | 0.49 |
| Complete | Euclidean | 0.99 | 0.99 | 0.49 | 0.49 |
| Single | Euclidean | 0.99 | 0.99 | 0.49 | 0.49 |
| Average | Euclidean | 0.99 | 0.99 | 0.49 | 0.49 |
| Ward | Euclidean | 0.99 | 0.99 | 0.49 | 0.49 |

## I.2 Edge cases

In the previous sub-section, we explored different metrics for determining the optimal number of clusters in the Agglomerative clustering algorithm. However, three out of the four metrics are not applicable when the number of clusters is equal to the number of units or when there is only one cluster. Unfortunately, even the modified modularity metric fails to detect these cases accurately.

Let $\tilde{D}$ be the normalized distance matrix of the unit feature vectors, where the sum of all elements is equal to $1$. Consider the units partitioned into $k$ clusters and a matrix $A \in \mathbb{R}^{k \times k}$, where $A_{ij} = \sum_{a \in C_i, b \in C_j} \tilde{D}_{ab}$ represents the sum of the distances between all pairs of units in clusters $C_i$ and $C_j$. The modularity metric, denoted by $M$, measures the quality of the partitioning and is defined as:

$$M = \sum_{i=1}^{k} \left( A_{ii} - \left[ \sum_{j=1}^{k} A_{ij} \right]^2 \right) \quad (3)$$

The first term in this equation measures the total intra-cluster distance between the unit features while the second term is the expected intra-cluster distance under the null hypothesis that unit distances were randomly assigned based on joint probabilities in $\tilde{D}$. A negative value of $M$ indicates that pair-wise distance within a cluster is lower than the random baseline.

**Graph structures for edge cases:** The units within a given layer must be in the same cluster if the majority of their outgoing connections lead to the same units in the subsequent layers. That is, if the feature vectors are dense or if they are sparse but have low pairwise distances. On the other hand, the units must belong separate clusters if their outgoing connections predominantly target different units in the later layers. That is, if the unit feature vectors are sparse and have high pairwise cosine distances. Surprisingly, despite being at opposite ends of the clustering spectrum, these two cases can exhibit highly similar graph structures. In fact, both structures may closely resemble randomly shuffled graphs with the same number of edges as the original graph.

**Behavior of the modularity metric:** The modularity metric often fails to distinguish between the previously discussed graph structures and struggles to detect them accurately. Specifically, when there is only a single cluster ($k = 1$), the modularity metric always yields a value of $0$, irrespective of the pairwise cosine distances. Similarly, when $k = N$, the modularity metric values ($M$) are always less than or equal to $0$, regardless of the pairwise cosine distances. Consequently, even a slightly sparse random graph can result in negative modularity metric values for $k = N$ (as well as other $k$ values).

For instance, consider the function graph, neural network, and modularity metric values depicted in Figure 38. In the input layer, the connectivity or unit feature vectors exhibit slight sparsity. Despite the clear expectation that all input units should belong to a single cluster, the modularity metric value is the lowest for $k = 4 = N$.

Figure 38: Row 1: a. Function graph with a reused sub-function, b. hierarchically modular structure detected through a NN with 3 hidden layers and width of 24. ;Row 2: Modularity metric values for various layers and varying number of clusters $k$

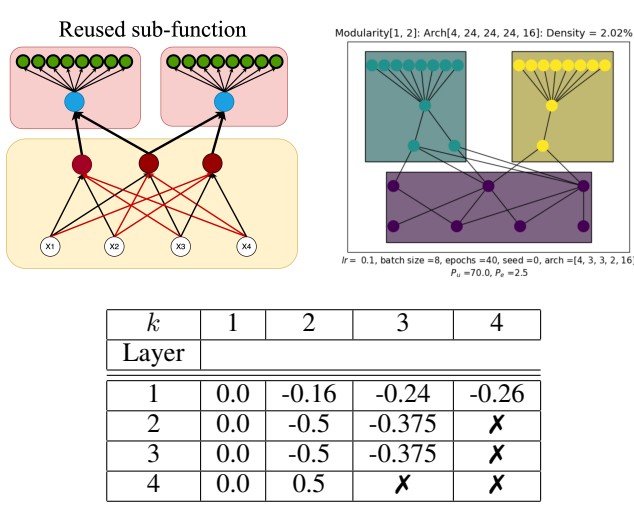

| $k$ | 1 | 2 | 3 | 4 |
|---|---|---|---|---|
| Layer | | | | |
| 1 | 0.0 | -0.16 | -0.24 | -0.26 |
| 2 | 0.0 | -0.5 | -0.375 | ✗ |
| 3 | 0.0 | -0.5 | -0.375 | ✗ |
| 4 | 0.0 | 0.5 | ✗ | ✗ |

Figure 39: Row 1: a. Function graph with 3 input separable and reused sub-functions, b. hierarchically modular structure detected through a NN with 3 hidden layers and width of 24. ;Row 2: Modularity metric values for various layers and varying number of clusters $k$

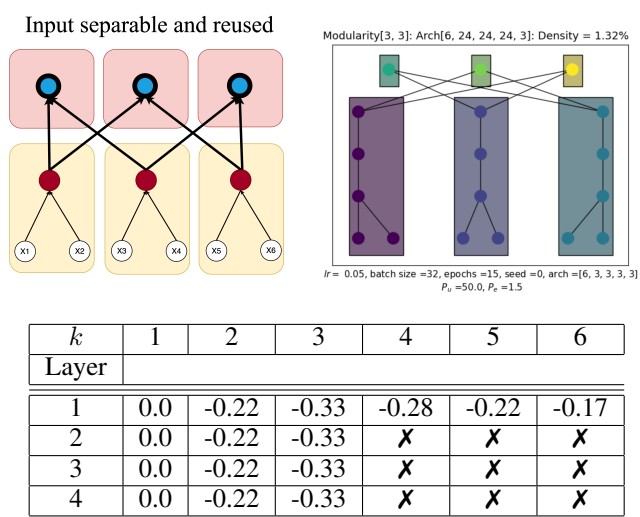

| $k$ | 1 | 2 | 3 | 4 | 5 | 6 |
|---|---|---|---|---|---|---|
| Layer | | | | | | |
| 1 | 0.0 | -0.22 | -0.33 | -0.28 | -0.22 | -0.17 |
| 2 | 0.0 | -0.22 | -0.33 | ✗ | ✗ | ✗ |
| 3 | 0.0 | -0.22 | -0.33 | ✗ | ✗ | ✗ |
| 4 | 0.0 | -0.22 | -0.33 | ✗ | ✗ | ✗ |

Figure 40: Row 1: a. Function graph with a single densely connected sub-function, b. hierarchically modular structure detected through a NN with 3 hidden layers and width of 24. ;Row 2: Modularity metric values for various layers and varying number of clusters $k$

Single sub-function

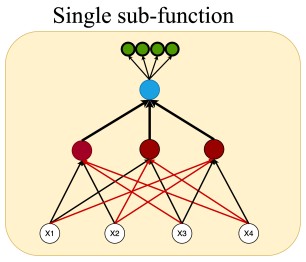
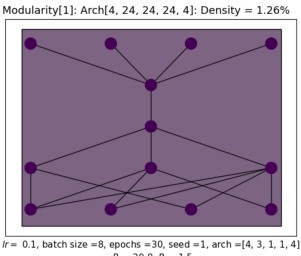

| $k$ | 1 | 2 | 3 | 4 |
|---|---|---|---|---|
| Layer | | | | |
| 1 | 0.0 | 0.0 | 0.0 | 0.0 |
| 2 | 0.0 | 0.0 | 0.0 | ✗ |
| 3 | 0.0 | ✗ | ✗ | ✗ |
| 4 | 0.0 | ✗ | ✗ | ✗ |

In Figure 39, we can observe that in layers 2, 3, and 4, the modularity metric value is lowest for $k = 3 = N$, which aligns with the ground truth. However, due to our previous observation regarding $k = N$ consistently yielding the lowest modularity metric value, regardless of the ground truth being $k = 1$, this becomes an unreliable outcome. The fact that $k = N$ always generates negative modularity metric values undermines its effectiveness in detecting the case where $k = N$.

Lastly, Figure 40 illustrates that if the neural network is densely connected, the modularity metric values for all $k$ are 0, further complicating the search for an exact solution. As a result of these observations, we have developed an additional tool to identify the edge cases of $k = 1$ and $k = N$.

**Unit separability test:** The unit separability test is designed to evaluate whether two units in a cluster can be separated into sub-clusters. Consider two units $i$ and $j$, with $o_i = \sum f_i$, $o_j = \sum f_j$ neighbors respectively, and $o_{ij} = f_i \odot f_j$ common neighbors. We consider a random baseline that preserves $o_i$ and $o_j$. The number of common neighbors is modeled as a binomial random variable with $g$ trials and probability of success $p = \frac{o_i \times o_j}{g^2}$, where $g$ is the total number of units in the later layer. The units are separable if the observed value of $o_{ij}$ is less than the expected value $\mathbb{E}(o_{ij})$ under the random model.

We can observe in the previous examples that if those edge cases exist in the ground truth, usually $k = 2$ or $k = N$ has the lowest modularity metric among $k = 2, ..., N$.

**Test 1:** Consider the partition where $N - 1$ clusters are obtained. If the two units that are found in the same cluster are separable, it implies that all units belong to separate clusters.

**Test 2:** Let us consider the partition of units into two clusters. We merge the feature vectors of the two unit groups. If the two groups of units are not separable, it implies that all units must belong to the same cluster.

In some cases, both tests yield positive results as the graph structures are similar (discussed above). We determine the optimal number of clusters by selecting the result that is more statistically significant. The detailed pseudo code for unit clustering is shown in algorithm 3.

