# OpenReview forum: "Neural Sculpting: Uncovering hierarchically modular task structure in neural networks through pruning and network analysis"
_NeurIPS.cc/2023/Conference — NeurIPS 2023 poster_

### Official Review · Reviewer_znid · 2023-07-02

**Soundness:** 2 fair
**Presentation:** 3 good
**Contribution:** 2 fair
**Rating:** 6
**Confidence:** 4

**Summary:**

When conventionally trained, neural networks do not demonstrate structural properties like input separable functions or reusability of sub-modules. The authors investigated this phenomenon and proposed iterative pruning to enhance structural properties of neural networks. They demonstrate the effectiveness of their method on boolean functions and a modified MNIST dataset.

**Strengths:**

* The paper is well-written with nice motivations and clear logical flow.
* The paper investigates this interesting scientific question "whether conventionally trained neural networks display structural properties", which is an important question for network interpretability
* The paper proposes iterative pruning to enhance structural properties, which is a technical contribution.

**Weaknesses:**

* The scope is a bit limited. The paper only discusses two properties, input separable functions and reuse of sub-modules. The examples are a bit too simple. Said so, maybe this is not too much of a problem for a scientific paper whose goal is to understand something. Still, I'd love to see your method applied to larger-scale experiments.
* The novelty is not very clear to me. I'm glad to see that iterative pruning works pretty well for module reuse, but it seems there is not too many technical contribution. I suggest authors should highlight the technical contributions/comparisons to previous works.
* Regarding the method, I feel encouraging modularity directly in training (e.g., adding Eq. (2) as a penalty in training) may further enhance structural properties. I'd love to see how this trick changes the outcomes (especially for the failed cases).

**Questions:**

* Figure 3 and Figure 4 seem to have the exact same titles and captions. Should distinguish them more clearly.
* I'm not very sure about your definition of reusability (your definition sounds more like "shared features" to me). An example I would count as reuse of submodule is: consider input (x1, x2), output ((x1-x2)^2, (x1+x2)^2), the squared function should be learned twice (reused). However, there is probably no shared feature in a trained network. A trained network (e.g., a fully-connected network) can only learn the squared function twice and independently, even with your pruning strategy I guess.
* Do you expect your method to generalize to larger models, e.g., large language models?

---

> ### Author Rebuttal · Authors · 2023-08-09
>
> We thank the reviewer for their constructive comments. We provide responses to the identified weaknesses and questions raised next:
>
> **W1: The scope is a bit limited. The paper only discusses two properties, input separable functions and reuse of sub-modules. The examples are a bit too simple.This is not too much of a problem for a scientific paper whose goal is to understand something.**
>
> As the reviewer also pointed out, the goal of the paper is to understand under which training constraints neural networks (NNs) acquire the hierarchically modular structure reflecting that of the given task – and it proposes a methodology to uncover that structure. The use of simple Boolean function graphs and sub-function properties allows for a principled analysis, making it easier to gain insights into the resulting NN structure. Further, the evaluation of the proposed training and network analysis methodology requires the knowledge of the task’s structure (ground truth). Boolean functions allow us to construct a wide range of tasks with different (yet known) hierarchical structures for detailed evaluation as demonstrated in section 5. Whether properties other than input separability and module reuse are similarly discoverable through our proposed methodology remains to be investigated in future work. However, we believe that these two properties are both quite central and general.
>
> **W2: The novelty is not very clear to me. I'm glad to see that iterative pruning works pretty well for module reuse, but it seems there is not too many technical contributions. I suggest authors should highlight the technical contributions/comparisons to previous works.**
>
> We would like to highlight the technical contributions in our work:
>
> 1. Propose a novel training methodology based on iterative pruning of units and then edges that results in NNs with a hierarchically modular structure that reflects the corresponding structure of the given task.
> 2. Develop a method based solely on unit connectivity to organize units remaining after pruning into modules and infer the hierarchical structure learned. The method utilizes path-based unit features, clustering, and cluster merging to uncover the underlying hierarchy of modules.
> 3. Design experiments and analysis tools to detect whether NNs, after training, acquire structural properties resembling the properties of the task’s sub-functions.
> 4. While individual components of our proposed training (pruning) and module detection methodology have been previously explored, our main contribution is to synthesize these components into a coherent and empirically evaluated pipeline.
>
> To the best of our knowledge, our paper is the first work that presents a combined training (pruning) and network analysis tool (module detection) to uncover hierarchical modularity. Previous works in modularity and NNs have only worked on the latter. Due to page limitations, we combined comparisons with previous works in the introduction section (paragraph 2). That paragraph provides a high level overview of prior work and differences with our paper. We will include a more detailed comparison at a technical level in the camera-ready version, using the additional page provided.
>
> **W3: I feel encouraging modularity directly in training (e.g., adding Eq. (2) as a penalty in training) may further enhance structural properties.**
>
> Thank you for this very interesting suggestion. The concept of hierarchical modularity naturally incorporates sparsity and reusability, resulting in more efficient task/function representations. Our pruning approach revolves around the idea of first restricting the number of units to promote module reuse and then the number of edges to reveal the sparse connectivity, thus promoting the emergence of hierarchical modularity.
>
> Adding a penalty term to the loss function to restrict the number of units and edges is a promising idea. However, it requires careful consideration, including the sequential nature of unit and edge penalty application (refer to section 3) to capture densely connected reused modules effectively. We acknowledge the potential of this approach to reduce training costs. It is a future research direction that is definitely worth exploring.
>
> **Q1: Figures 3 and 4 seem to have the exact same titles and captions.**
>
> We will make the figure titles and captions more distinguishable in the final version.
>
> **Q2: I'm not very sure about your definition of reusability. An example I would count as reuse is: consider input (x1, x2), output ((x1-x2)^2, (x1+x2)^2), the squared function should be reused. A trained network can only learn the squared function twice and independently, even with your pruning strategy.**
>
> We want to clarify the distinction between function reuse and operation reuse, as we define them in this work. In the example provided, squaring is an operation that is independent of its input variables. On the other hand, a function is a combination of such operations along with specific ordered input variables to it. As correctly pointed out, specific operations have to be relearned if applied to different inputs due to the fixed data flow in NNs. In systems with dynamic routing, it would be possible to learn squaring only once, qualifying as a reused operation. However, in fixed graphs like NNs, this is not feasible, as also highlighted in prior works (Gref et. al. 2020, Csordas et. al. 2021). We will clarify this distinction and elaborate on it in the next version of the paper.
>
> **Q3: Do you expect your method to generalize to larger models?**
>
> We expect the overall idea of restricting the number of units and edges (see W3) to remain valid. We do hope to apply our method to larger models and analyze the resulting structure. However, this remains out of scope for this work as the hierarchical structure of those tasks is unknown (see W1) and any identified structure couldn’t be compared against a ground truth.

---

> > ### Comment · Reviewer_znid · 2023-08-10
> >
> > Thanks for clarification! I will raise my score from 5 to 6.

---

### Official Review · Reviewer_fXf6 · 2023-07-05

**Soundness:** 4 excellent
**Presentation:** 4 excellent
**Contribution:** 4 excellent
**Rating:** 8
**Confidence:** 5

**Summary:**

This paper conducts an investigation of hierarchical modularity in neural networks by studying boolean networks. The paper studies simple hierarchically modular boolean functions and learns them using MLPs. It then propose metrics to discover this modularity by examining 1) input separability and 2) reusability of sub-functions. Finally, the paper provides a clustering based method to identify the modules in neural networks applied to arbitrary tasks.

**Strengths:**

1) Clear and scientifically principled investigation into an important yet understudied facet of neural networks

2) Interesting results showing that hierarchical modularity in neural networks often doesn't emerge with standard training but with pruning (both edge and neuron), the sparsity forces the networks to become hierarchically modular.

3) Novel method to find  with some empirical backing

**Weaknesses:**

1) The experiments on more realistic datasets e.g. MNIST can be expanded to link the findings of the paper more concretely to what is observed in practice.



**Questions:**

Currently, the experiments on MNIST to verify the method to discover modularity do serve as a proof of concept but more extensive experiments would be needed to confirm the effectiveness of the proposed approach to identify modules in neural networks in general.

In particular, extensions to CNNs on more realistic image datasets might be very interesting. It might be useful to conduct experiments with a subset of classes from a dataset like CIFAR100 or ImageNet where the superclasses and subclasses often offer some natural opportunities for hierarchical modularity.

---

> ### Author Rebuttal · Authors · 2023-08-09
>
> We thank the reviewer for the constructive comments about this work.
>
> **Weakness: Currently, the experiments on MNIST to verify the method to discover modularity do serve as a proof of concept but more extensive experiments would be needed to confirm the effectiveness of the proposed approach to identify modules in neural networks in general. In particular, extensions to CNNs on more realistic image datasets might be very interesting. It might be useful to conduct experiments with a subset of classes from a dataset like CIFAR100 or ImageNet where the superclasses and subclasses often offer some natural opportunities for hierarchical modularity**
>
> The primary aim of this paper is to understand the conditions under which a neural network will acquire a hierarchically modular structure that reflects the functional structure of the given task – and to design a methodology for uncovering that structure. To validate this approach, tasks with known hierarchical structures are needed. Boolean functions offered a diverse range of tasks with distinct structures, allowing for an effective evaluation of the proposed methodology. Additionally, as the reviewer points out, the MNIST experiments provide a proof of concept for relatively larger-scale applicability.
>
> While we acknowledge the potential for broader experiments involving larger models and complex tasks like CNNs or transformers, that would be a highly intriguing future research direction that is naturally a follow-up work to this first study.
>
> We hope this perspective resonates with the reviewer's understanding.

---

> > ### Comment · Reviewer_fXf6 · 2023-08-10
> >
> > I have read the response and I stand by my original assessment of the paper.

---

### Official Review · Reviewer_Czk6 · 2023-07-09

**Soundness:** 4 excellent
**Presentation:** 3 good
**Contribution:** 3 good
**Rating:** 7
**Confidence:** 3

**Summary:**

The paper proposes a methodology for uncovering hierarchical modularity in neural networks (NNs). It combines iterative pruning and network analysis to reveal the underlying hierarchy of sub-functions in tasks. The paper demonstrates the effectiveness of the method on both Boolean functions and vision tasks using the MNIST dataset.

The main contribution of the paper lies in providing a novel approach to uncover hierarchical modularity without prior knowledge of the task's hierarchy.

The methodology offers insights into efficient and interpretable learning systems and showcases the potential of pruning and network analysis methods in revealing and utilizing structural properties in NNs.

**Strengths:**

The paper demonstrates a significant strength through its comprehensive experimental evaluation, encompassing modular and hierarchical Boolean function graphs, as well as tasks utilizing the MNIST digits dataset. The authors meticulously conduct numerous trials, systematically varying network parameters like depth, width, and seed values to thoroughly validate the efficacy of their proposed methodology. The experimental results substantiate the approach's ability to precisely uncover the hierarchical and modular structures within the tasks, serving as empirical evidence of the methodology's robustness and broad applicability.

**Weaknesses:**

One potential weakness of the paper is the limited discussion and analysis of the results regarding the failures or limitations of the proposed methodology. While the experiments highlight the success rates in detecting modules and uncovering hierarchical structures, there is less exploration of cases where the methodology might not perform as well or situations where the detected modules do not align perfectly with the expected sub-functions. A more in-depth analysis of the challenges and limitations of the approach could provide valuable insights for further improvement and understanding.

**Questions:**

1. How does the proposed iterative pruning process affect the overall performance of the neural networks in terms of learning efficiency and accuracy?
2. Are there any limitations or challenges encountered when applying the proposed methodology to more complex tasks beyond Boolean functions and MNIST digit classification?
3. Could the approach be extended to tasks with dynamic or evolving hierarchical structures, where the sub-functions change over time?
4. How does the proposed methodology compare to existing approaches in terms of accuracy, efficiency, and scalability when uncovering hierarchical modularity in neural networks?

**Limitations:**

Minor comments:
- Figure 5B1 -> the text is hard to read on a printed paper, due to small fonts in the figure. This is a minor comment, because if one is reading the PDF paperlessly, one can zoom in.

---

> ### Author Rebuttal · Authors · 2023-08-09
>
> We thank the reviewer for their constructive comments and appreciate your valuable feedback. Below we provide response to the weaknesses and questions:
>
> **W1: One potential weakness of the paper is the limited discussion and analysis of the results regarding the failures or limitations of the proposed methodology.**
>
> Please note that the paper clearly identifies some failure cases in the experiments section and appendix sections. However, we acknowledge the importance of providing more in-depth analysis of those cases. To improve the paper, we will extend the appendix sections and update the experiments section in the camera-ready version, offering a more comprehensive examination of the limitations and challenges encountered.
>
> **Q1: How does the proposed iterative pruning process affect the overall performance of the neural networks in terms of learning efficiency and accuracy?**
>
> The iterative pruning process, combined with cyclic learning rates, is computationally demanding compared to other pruning methods. During each iteration of the pruning process, units and edges are eliminated, which reduces the operation count for each iteration, still the overall cost is larger than training the dense neural network (NN). However, iterative pruning has been shown to produce highly sparse NNs that generalize well compared to other pruning methods (Leslie N Smith et. al. 2017, Alex Renda et. al. 2020). Our objective is to constrain the NNs to utilize as few units and edges as possible while still learning the task well. Further, the algorithm must operate without any prior knowledge of the final NN configurations (width, density). The iterative pruning algorithm is well-suited for this purpose. Improving computational efficiency is a future research direction that is worth exploring. (additionally see response to reviewer znid, W3)
>
> Throughout the pruning process, the validation accuracy is consistently maintained at the same level as the dense NN. When the NN can no longer achieve the desired accuracy, the pruning process is halted, and the algorithm reverts to the previous sparse NN. Despite pruning, the test accuracy remains largely unchanged, primarily due to longer training time (Tian Jin et. al. 2022) and the structure learned by the NNs.
>
> **Q2: Are there any limitations or challenges encountered when applying the proposed methodology to more complex tasks beyond Boolean functions and MNIST digit classification?**
>
> We focus on Boolean tasks because it is easier to know their correct hierarchical structure (ground truth), which is required for evaluating our methodology. Analyzing the resulting structure of larger models on more complex tasks could be a follow up work.
>
> For larger networks and complex tasks, it’s possible that NNs may learn numerous functional decompositions for the same task. The proposed pipeline can uncover only one of these decompositions, which could significantly differ from other possible ones.
>
> We anticipate good structures to be extracted for tasks learned using MLP-based NNs (e.g., simple MLPs, transformers). For CNNs, adapting unit pruning and clustering features for convolutional layers may also pose additional challenges but the methodology is expected to work well for CNNs after appropriate adjustments.
>
> **Q3: Could the approach be extended to tasks with dynamic or evolving hierarchical structures, where the sub-functions change over time?**
>
> If our interpretation of this question deviates from your intended one, please let us know. In scenarios where tasks vary, leading to evolving hierarchical structures and sub-functions over time, our current method may not be directly applicable. Our approach requires NNs to first learn a task well before pruning, making it less suitable for dynamic task settings. However, exploring this direction in future research could be promising. Limiting the number of units and edges under evolving tasks may naturally promote the reuse of sub-functions (Kashtan et. al. 2007). Dynamic sparse training algorithms (Mostafa et. al. 2019, Evci et. al. 2019) with growing and pruning of edges during training may facilitate such adaptability.
>
> **Q4: How does the proposed methodology compare to existing approaches in terms of accuracy, efficiency, and scalability when uncovering hierarchical modularity in neural networks?**
>
> To the best of our knowledge, this work is the first to propose a combined training (pruning) and network analysis tool to uncover hierarchical modularity. However, previous works have proposed methods to detect modules in trained NNs without requiring knowledge of sub-functions or data (structural decompositions).
>
> Daniel Filan, Shlomi Hod, and colleagues (Filan et al. 2021, Hod et al. 2022, Casper et al. 2022) employed normalized spectral clustering to globally extract unit clusters and analyze them. Spectral clustering optimizes for N-cuts, measuring internal connectivity against external connectivity of unit clusters. Our experiments with that method suggest that it often does not uncover the expected modules in sparse NNs. This could be attributed to its global nature and the absence of edges between NN units at the same layer.
>
> Watanabe and colleagues (Watanabe et al., 2018; Watanabe, 2019) published a sequence of interesting papers where they utilized layer-wise clustering of units based on incoming and outgoing connectivity. Our method aligns with this class of previous methods. Although we have not directly tested the latter on the pruned NNs, it’s worth noting that they were designed for conventionally trained NNs. In contrast, our method is simpler and more tailored to the pruned NNs we obtain. Due to the limited time available we were unable to make such quantitative comparisons during the rebuttal phase.

---

> > ### Comment · Reviewer_Czk6 · 2023-08-18
> >
> > Thank you for the clarification. I have read the response.

---

### Author Rebuttal · Authors · 2023-08-09

We would like to thank all reviewers for their comments, time and effort. We have carefully thought about and responded to individual reviews, focusing on the weaknesses pointed out and the questions asked. If additional details, explanations, or clarifications are needed, we will be happy to provide them.

---

### Decision · Program_Chairs · 2023-09-21

**Decision:**

Accept (poster)

**Comment:**

This paper proposes a new method for uncovering hierarchical modularity in neural networks. It combines iterative pruning and network analysis to reveal the underlying hierarchy of sub-functions in tasks. Experiments are conducted on Boolean functions and a modified MNIST dataset.

After rebuttal, it received scores of 678. All the reviewers agree that (1) this paper provides a novel method to uncover hierarchical modularity without prior knowledge of the task's hierarchy. It provides clear and scientifically principled investigation into an important yet understudied facet of neural networks. (2) Experiments are convincing, which show the potential of pruning and network analysis methods to force neural networks to become hierarchically modular. On the other hand, a shared concern by the three reviewers is that the experiments on MNIST is more like a proof of concept. Because of this, I cannot push this accept to a spotlight. Analyzing the resulting structure of larger models on more complex tasks could be a very interesting follow up work.

Overall, given the general positive review feedback, the AC would like to recommend acceptance of the paper.